# A memory transcriptome time course reveals essential long-term memory transcription factors

Spencer G. Jones [1,9], Beatriz Gil-Martí [2,3,9], Eva Sacristán-Horcajada[4], Abigail C. Edison[1], Emily F. Butler [1], Neda Miandashti[1], Camilla Roselli [5,6], Enrique Turiégano[3], Tamara Boto [6,7], Jamie M. Kramer [1,9] ✉ & Francisco A. Martin[2,8,9] ✉

Long-term memory (LTM) requires transcription and translation of new proteins, yet the transcriptional control of memory remains poorly understood. Here, we performed a transcriptome time-course during LTM formation in *Drosophila melanogaster* exposed to courtship conditioning. We identified a mushroom body-specific transcriptional memory trace that becomes activated during memory consolidation. Using scRNAseq of CREB-activated cells we were able to detect a persistent transcriptional response in MB neurons after LTM consolidation and retrieval. As a proof of causality, we conducted a loss-of-function screen for genes comprising the transcriptional memory trace, finding 16 positive hits whose disruption impaired LTM. Among them, we identified two neuron activity-regulated genes, *Hr38* and *sr*, which encode transcription factors that are activated by courtship LTM training, required for LTM, and bind to many genes comprising the transcriptional memory trace. Overall, we further define the transcriptional response to LTM and identify transcription factors that may help shape it.

Across several different species and memory paradigms, it has been shown that consolidation of short-term memory (STM) into long-term memory (LTM) requires activated gene transcription and translation of new proteins[1–3]. Early work in sea slugs and flies implicated cAMP signaling and the cAMP response element binding protein (CREB) in the formation of stable memories[4–10]. CREB is a transcription factor that is thought to trigger a wave of neuron activity-regulated genes (ARGs), which includes immediate-early genes (IEGs), activated at the onset of memory formation[11,12]. These IEGs in turn are predicted to regulate the expression of downstream secondary response genes that help mediate LTM formation[13,14].

*Drosophila melanogaster* has long been used as a model to study the molecular mechanisms underlying memory. The mushroom body (MB) is the memory center of the fly brain[15–18]. MB ablation eliminates different types of learning and memory, without a drastic effect on other behaviors[17,19]. The MB is composed of three different types of Kenyon cells, the α/β, α'/β', and γ neurons[18]. Several studies have shown that blocking translation, or disrupting CrebB transcriptional activity in the MB, or MB output neurons, leads to loss of LTM[20–24]. This suggests that transcription is essential for memory formation in flies, but our understanding of the memory-induced transcriptome is quite limited. In contrast to mammalian systems, where memory IEGs such

¹Biochemistry and Molecular Biology, Dalhousie University, Halifax, NS, Canada. ²Cajal Institute, Spanish National Research Council (CSIC), Madrid, Spain. ³Department of Biology, Autonomous University of Madrid, Madrid, Spain. ⁴Biocomputational Analysis Core Facility, Centro de Biología Molecular Severo Ochoa, Spanish National Research Council (CSIC), Madrid, Spain. ⁵Department of Genetics and Microbiology, Trinity College Dublin, Dublin 2, Ireland. ⁶Trinity College Institute of Neuroscience, Trinity College Dublin, Dublin 2, Ireland. ⁷School of Physiology Pharmacology and Neuroscience, University of Bristol, Bristol, UK. ⁸Department of Biochemistry and Molecular Biology, Faculty of Biological Sciences, Complutense University of Madrid, Madrid, Spain. ⁹These authors contributed equally: Spencer G. Jones, Beatriz Gil-Martí, Jamie M. Kramer, Francisco A. Martin. ✉e-mail: jkramer@dal.ca; fmarti22@ucm.es

as *Arc* and *cFos* are well defined[12], no IEG involved in memory formation in flies has been identified. Several studies have performed transcriptome profiling of whole fly heads or MB neurons after memory training, which resulted in the identification of some genes that are induced by memory and required for memory formation[25–32]. Despite this progress, there is little overlap observed between the different datasets. In addition, studies focused on detailed transcriptional profiling of individual experience-induced cells are scarce. Clearly, a more comprehensive analysis of the memory transcriptome is needed.

Here, we performed a time course analysis of the MB memory transcriptome, tracking gene expression during courtship conditioning, a complex learning and memory paradigm that is dependent on a recurrently activated circuit involving MB neurons[17,33–38]. In this assay, male flies are trained by prolonged exposure to a non-receptive mated female, who rejects copulation attempts. As a result, male flies learn to suppress courtship behavior during subsequent encounters with mated females[39,40]. We contrasted the MB training induced transcriptome to the whole head (WH) and identified a MB-specific transcriptional trace of courtship LTM that is active during training and LTM consolidation. We then employed scRNAseq to determine the persistent transcriptional response to LTM formation in a subset of MB neurons with activated CREB. Among the genes induced by courtship LTM training, we performed functional testing to identify 16 genes required for LTM, including two candidate memory ARGs, *Hormone receptor-like in 38* (*Hr38*) and *stripe* (*sr*). Overall, this study reveals key processes that are transcriptionally induced in the MB during courtship conditioning and identifies key transcription factors that may shape the MB transcriptional trace of LTM.

## Results

### A MB-specific transcriptional trace of courtship LTM

To better understand the role of gene activation in courtship LTM, we performed a transcriptome time course analysis of flies exposed to courtship LTM training and time-of-day matched naïve flies. To focus on a relevant memory cell type, we isolated nuclei from the MB using isolation of nuclei from a tagged cell type (INTACT)[27,41] and performed RNA-sequencing at eight time-points; three during the 7-h training period, and five after training (Fig. 1A). In parallel, we also sequenced RNA from WH nuclei. MB nuclei were tagged by expression of *5XUAS-unc84::GFP* with the *R14H06-Gal4* driver. As shown previously[27,42], this Gal4 line is highly specific for Kenyon cells (KCs), with only sparse and weak labeling of non-KCs (Fig. S1A). By comparing RNAseq data from WH and MB nuclei, we observed high enrichment of MB markers, and depletion of non-MB genes (Fig. S1B), suggesting that this protocol was effective at enriching for MB nuclei. To identify genes that were transcriptionally regulated in response to LTM training, differential expression analysis was performed for both MB and WH fractions by comparing trained flies with time-of-day matched naïve flies at six different time-points: 1 h and 7 h during training (DT), and 1 h, 7 h, 13 h and 19 h after training (AT) (Fig. 1A). Differentially expressed transcripts (FDR < 0.1) between naïve and trained conditions were classified as either training induced genes (TIGs) or training repressed genes (TRGs) if transcript levels increased or decreased, respectively, at any one of the time-points.

Overall, more genes were found to be differentially regulated in the WH fraction (2066) compared to the MB (1256) (Fig. 1B, Supplementary Data 1 and 2). Among TIGs, 68% were exclusively affected during training, while 32% showed some regulation after training. Most gene expression changes after training were identified in the early stages at 1hAT and 7hAT, with very few genes showing altered expression at 13hAT and 19hAT (Fig. 1B).

We next divided TIGs into three groups: WH-specific TIGs (*n* = 815), MB-specific TIGs (*n* = 333), and TIGs found in both the MB and WH (MBWH, *n* = 423) (Fig. 1C and Supplementary Data 3). For each group, we plotted the average expression level normalized to the naïve

baseline (Fig. 1D). As a group, WH-specific TIGs showed a strong induction above baseline near the end of the training period that was not observed in the MB. For MB-specific TIGs we observed high induction at the onset of training (1hDT), followed by reduced expression in the middle of training (3.5hDT) and reinduction at the end of training (7hDT) and 1 h after the end of training (1hAT) (Fig. 1D). In the MBWH group, we observed a strong induction over baseline at the onset of training in both MB and WH samples, but in the MB, these genes became reinduced again at the end of training and remained significantly more induced than in WH during the early post training period (Fig. 1D). Therefore, both the MB-specific and MBWH TIGs appear to represent a MB-specific transcriptional trace of courtship LTM, with activated expression specifically in the MB at the end of training, and post-training, when memory consolidation is thought to be occurring.

We next analyzed the functions of genes involved in the MB-specific courtship memory trace, including all TIGs in the MBWH and MB groups (*n* = 756). We conducted Gene Ontology (GO) enrichment analysis on MBWH + MB and the WH groups separately and looked for terms that were only enriched in MBWH + MB (Fig. 1E and Supplementary Data 3). This analysis revealed enriched GO terms that were relevant to the cellular corelates of LTM. For example, we observed enrichment of several actin cytoskeleton-related terms, including actin cytoskeleton organization (GO: Biological Process), which is important for synapse remodeling during memory formation[43]. TIGs annotated with this term showed clear increased induction late- and post-training in MB compared to WH samples (Fig. 1F). Individual actin-related genes show different dynamics of induction, for example the filamin ortholog *cheerio* is highly induced for the duration of training, while the actin binding protein *Arpc2* is only induced 1hAT (Fig. S2).

MB-specific energy metabolism via the TCA cycle is crucial for LTM, and not STM, in multiple *Drosophila* memory paradigms, including courtship LTM[44] and aversive olfactory LTM[45]. After memory training the MB goes through a period of high pyruvate flux through the TCA. Increasing the ability of MB neurons to process pyruvate through the TCA allowed LTM consolidation to occur more easily, suggesting that ATP generation is a critical trigger for LTM formation[45]. Interestingly, we found enrichment of terms related to these processes in the MBWH + MB gene set, including aerobic respiration (GO: Biological Processes) (Fig. 1E). These genes showed prolonged increased induction during the late training and post training phase of courtship conditioning in the MB compared to WH (Fig. 1G). This can be observed in individual gene plots, for example for the TCA cycle component mitochondrial aconitase 1 (mAcon1) and the electron transport chain component succinate dehydrogenase, subunit A (SdhA) (Fig. S2). These genes were induced over naïve at the start of training in MB and WH. Thereafter, expression was reduced in the WH to below naïve levels and reinduced in the MB. This observation correlates with the increased metabolic demand the MB experiences during memory consolidation[45].

Taken together, our courtship LTM transcriptome time course analysis reveals a MB-specific transcriptional memory trace in the late stages of training and the early post-training period when memory consolidation is occurring. This memory trace is enriched for genes involved in the cellular correlates of LTM. Many of these genes are broadly activated in the MB and WH during the early training stages, with MB-specific re-induction occurring later towards the end of training, and post-training.

### Post-training requirement of translation for courtship LTM

Many components of the translation machinery were identified among TIGs and TRGs in both WH and MB (Supplementary Data 1 and 2). Since translation is essential for LTM, we examined the expression of all genes annotated for "ribosome" (GO: Cellular Component) and "translation factors" (WikiPathways). In naïve MBs, these genes

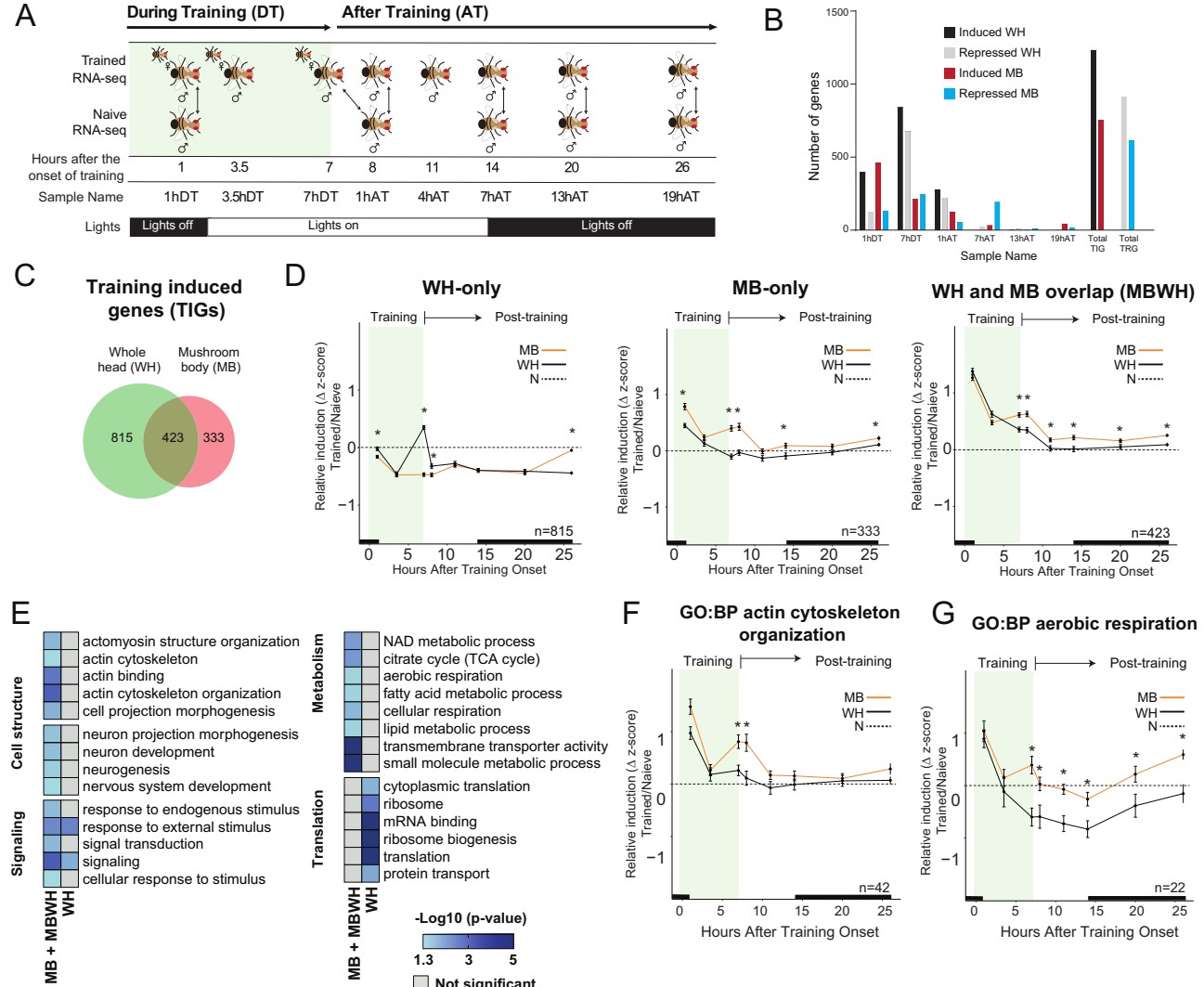

**Fig. 1 | Time course of courtship LTM shows a MB-specific transcriptional trace.** **A** Schematic of courtship training and sample collection approach for both trained and naïve flies. Arrows represent trained-naïve comparisons used in downstream differential expression analysis. **B** Differential expression analysis results, separated by upregulation or downregulation and by mushroom body (MB) or whole head (WH), for each trained-naïve comparison (FDR < 0.1). **C** Venn diagram of training induced genes (TIGs) identified in WH and MB. **D** Relative z-score induction during LTM formation in MB (orange) or WH (black) above naïve (N) baseline (dashed line) for TIGs identified only in the WH, only in the MB, or in both the WH and MB (MBWH). Statistical comparison between MB and WH was performed with a two-tailed t-test (*p < 0.05). The black bar indicates nighttime. **E** Gene ontology analysis

was performed for TIGs identified as part of the MB specific courtship memory trace (MB and MBWH, n = 756), as well as TIGs identified only in the WH (n = 815). A selection of significant (FDR < 0.05) curated metabolism, signaling, translation and cytoskeleton-associated terms are displayed. −Log₁₀ (p value) is represented in a heatmap. Relative z-score induction during LTM formation in MB or WH above naïve baseline for TIGs annotated to the selected significantly enriched GO terms **F** "actin cytoskeleton organization" and **G** "aerobic respiration". Statistical comparison between MB and WH was performed with a two-tailed t-test (*p < 0.05). The black bar indicates nighttime. Error bars indicate the standard error of the mean (SEM).

fluctuate in concordance with circadian activity rhythms showing a peak of expression precisely at the day to night transition and a low point of expression during the mid-day siesta (Fig. 2A). In trained flies, this pattern is completely disrupted. Translation genes are not reduced in expression during mid-day. Instead, they maintain a more consistent expression level during the day, while training is occurring, and after training during the consolidation period, with a peak in expression at 4hAT (Fig. 2A). Examples of expression profiles for specific translation initiation factors (eIF3b, eIF1A) and ribosomal proteins (RpS21, RpS9) are shown in Fig. S2. Translation is one of the essential processes that is crucial for LTM formation in many paradigms[4,6], but the requirement for translation in courtship LTM has not been previously tested. We inhibited protein synthesis by feeding wild type flies media containing cycloheximide (CXM) beginning 1 day prior to STM or LTM training, and tested memory ability. Flies fed CXM

were able to form STM, but not LTM (Fig. 2B–D). Flies that were removed from CXM media at the end of training, therefore allowing for translation to resume during the post training consolidation period, showed normal LTM (Fig. 2B–D). Taken together, this suggests that translation is required for courtship LTM, with translation occurring during the post-training period being the most critical for normal LTM formation.

## Altered expression of memory signaling genes during LTM formation
Previously, in two independent studies, we identified MB-specific induction of synaptic signaling genes in response to courtship LTM training at 1 h after the end of a 7 h training period[27,44]. We also observed enrichment of signaling genes among TIGs comprising the MB-specific memory trace (Fig. 1E). We compared TIGs identified at the

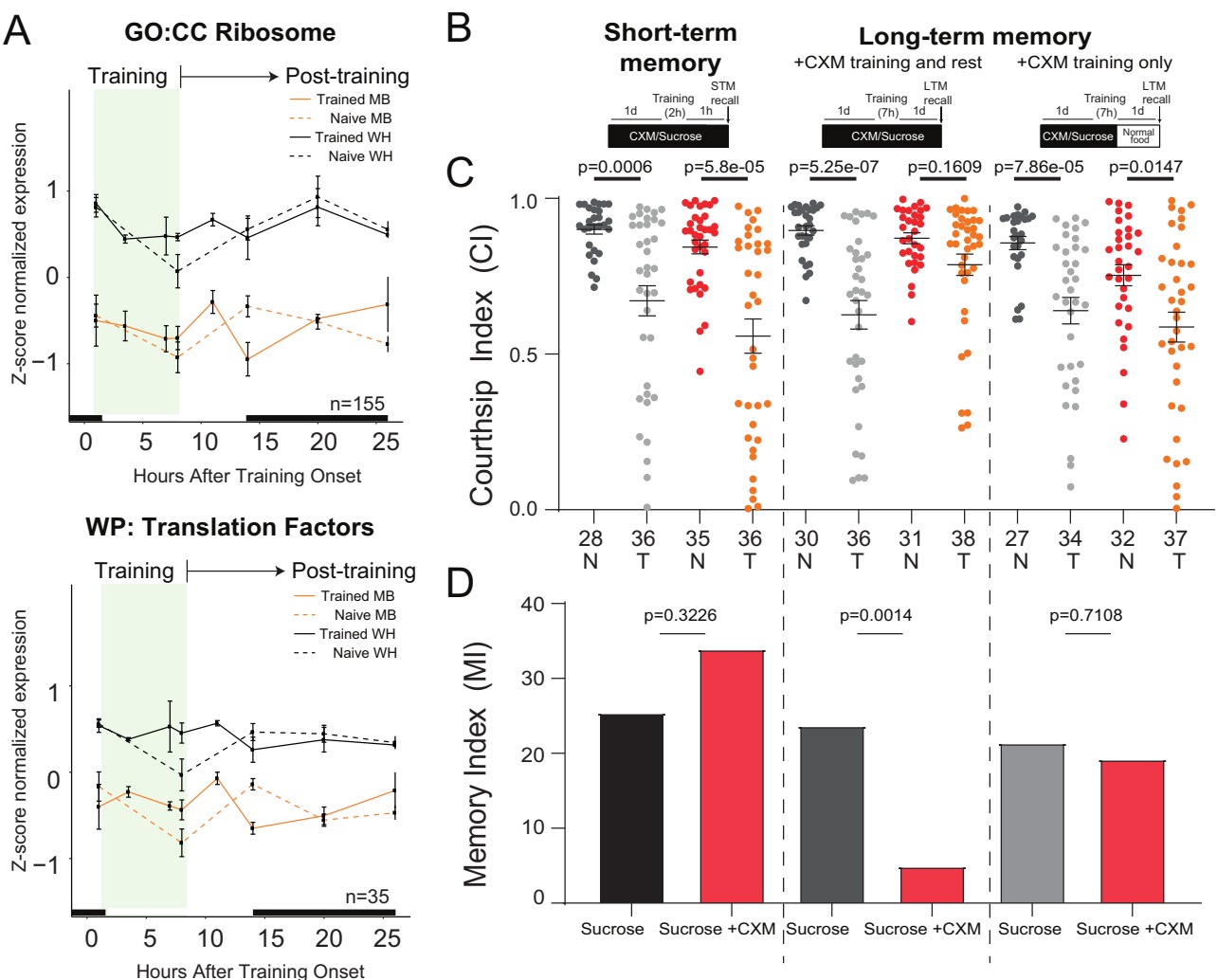

**Fig. 2 | Post training translation is required for courtship LTM. A** Mean z-score normalized expression in the MB or WH in naïve and trained flies of all expressed genes annotated to the GO:CC term ribosome (*n* = 155) or the WikiPathways term translation factors (*n* = 35). Black bars indicate nighttime. **B** Schematic diagrams of cycloheximide (CXM) feeding protocol. **C** Dot plot showing courtship indices (Cis) for naïve (N) and trained (T) flies. Statistical significance between naïve and trained flies was determined using a two-tailed Mann-Whitney test, *p* values are indicated. Number of flies tested for each condition is shown under corresponding dot plot. **D** Bar graph showing corresponding memory index (MI) derived from the CI (see "Methods"). *P* values were determined using a two-tailed randomization test with 10,000 bootstrap replicates. All error bars indicate the SEM. Source data are provided as a Source Data file.

end of training (7hDT) and 1 h after the end of training (1hAT) (Supplementary Data 2) and found significant overlap with the published datasets, including 68 core TIGs identified in at least 2 of 3 studies (Fig. 3A and Supplementary Data 4). As a group, these core TIGs show consistent significant induction across all three datasets (Fig. 3B) and are enriched for GO-terms related to memory, including synaptic signaling, G protein-coupled receptor signaling, and cAMP-mediated signaling (Fig. 3C). Interestingly, genes that are annotated for G protein- and cAMP-related signaling pathways fluctuate in naïve MBs in accordance with circadian activity rhythms (Fig. 3D). Like the translation genes (Fig. 2A), expression is high in naïve MBs at the day-night transitions when activity is high, and low in the middle of the dark or light period when activity is low (Fig. 3D). During LTM training these genes maintain their expression, which persists above naïve levels until the end of training or 1 h AT (Fig. 3D). A similar trend is observed in the WH data; however, the day-night cycling of these genes is not observed in WH, and the overall expression is much lower in WH than in the MB. Examples of expression profiles for cAMP signaling genes, including *Adenylate cyclase 1* (*Adcy1*, formerly known as *rutabaga*), *Dop1R2*, and *Pka-C1*, are shown (Fig. S2). Overall, these data suggest that courtship LTM training results in maintenance of memory signaling gene expression over time, in contrast to naïve flies, where expression levels fluctuate in the MB over time in a manner that correlates with daily activity rhythms and light.

## Isolation of persistent memory engram cells
The current view of memory formation is that experience triggers the activation of an ensemble of neurons, called an engram, which undergoes a period of consolidation and memory storage. The same stimulus presented again reactivates a subset of this ensemble and induces memory retrieval[46]. Our time course analysis identified broad transcriptional changes occurring during learning and memory consolidation in the MB but did not identify long lasting transcriptional changes that are expected in engram cells (Fig. 1B). To identify CREB activated neurons in the MB that might represent an engram of courtship memory, we used the CAMEL tool, a MB-specific transgenic construct that responds to activated CREB with the production of GFP[47] (Fig. 4A). The CAMEL tool mainly expresses in αβ and γ neurons, which is similar to the expression domain of *R14H06-Gal4* and includes the neuronal populations required for courtship LTM[35,36]. While CAMEL-positive non-MB neurons were detected by confocal microscopy, about 70% of CAMEL-positive cells were specifically MB cells

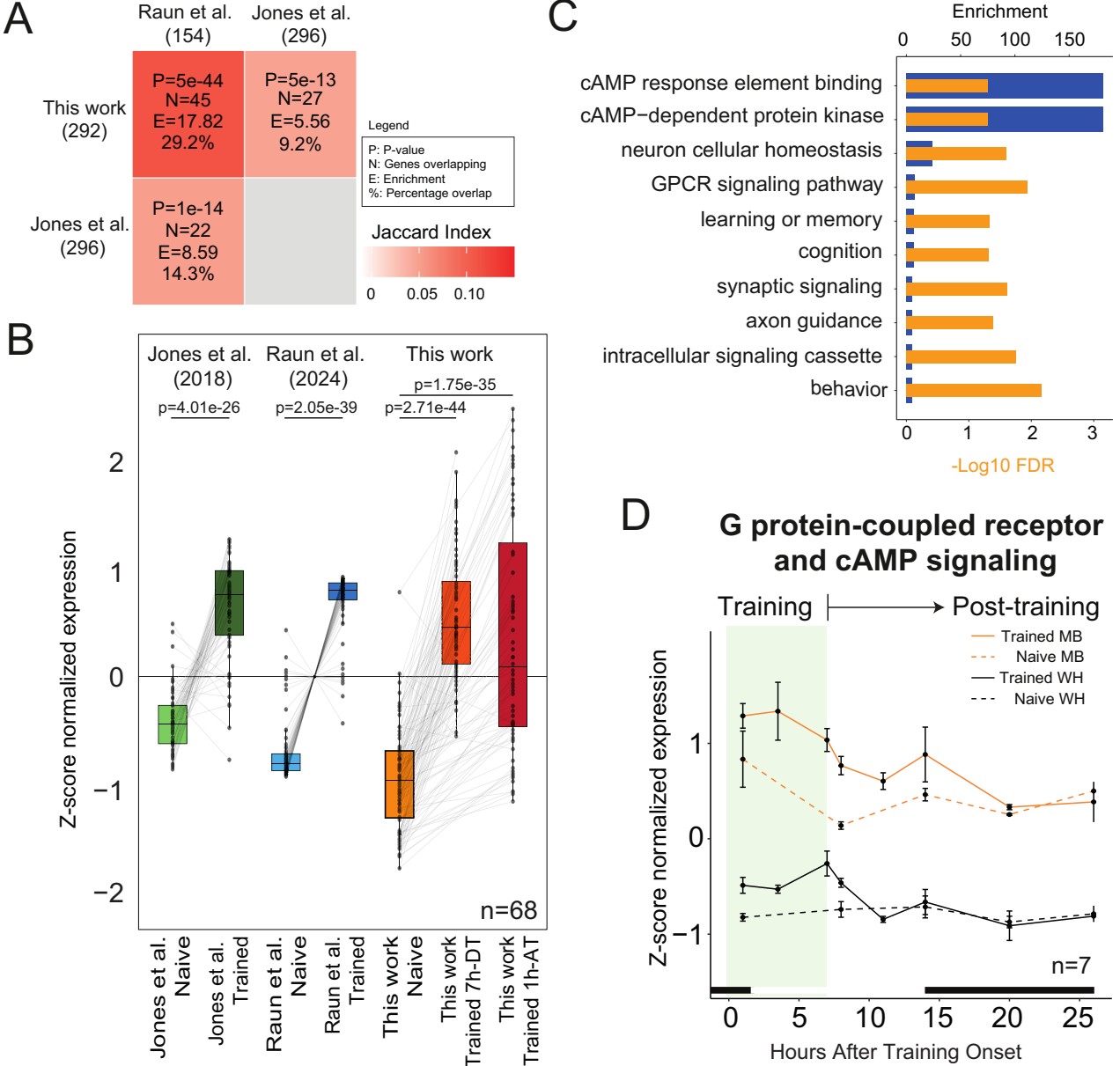

**Fig. 3 | Induction of memory signaling genes during LTM consolidation.**
**A** Heatmap indicating pairwise overlap between genes significantly induced in the MB at 7hDT or 1hAT in this time course with MB training-induced genes identified in other previously published datasets[27,44] Overlap statistics on each tile connecting two datasets indicate hypergeometric *p* value for the enrichment, the number of overlapping genes, fold enrichment, and percentage of overlap. **B** Boxplot displaying z-score normalized expression of 68 core TIGs found in two of three datasets. *P* values were calculated using a two-sided pairwise *t*-test. Boxes: 25–75th percentile; center line median; whiskers Tukey (±1.5 × Interquartile Range).
**C** Significant GO terms (FDR < 0.05) for the 68 core TIGs. **D** Mean z-score normalized expression of core TIGs annotated to GO terms related to G protein-coupled receptor signaling and/or cAMP signaling. Error bars indicate the SEM.

(Fig. S3). Different behavioral paradigms, such as olfactory aversive conditioning or social interaction, increased the number of CAMEL-positive cells in the MB after training[47,48]. We quantified the number of CAMEL-positive neurons 24 h after courtship LTM training. Trained flies showed a significant increase in the number of CAMEL positive MB neurons compared to naïve flies. In contrast, flies with a strong hypomorphic mutation in *Adcy1*[49] which disrupts LTM formation[3,9] did not display a change in the number of CAMEL-positive MB neurons between trained and naïve individuals (Fig. 4B). We confirmed the functional relevance of CAMEL-labeled neurons by expressing the neuronal silencing tool TNT[G] in CAMEL-positive neurons (*CAMEL>TNT[G]*), thus specifically blocking the neuronal outputs of CREB-induced MB neurons. At 24 h after training, we recorded and quantified the courtship behavior of *CAMEL>TNT[G]* males and controls

towards mated females. While control trained flies showed a reduction in their time of courtship when compared to control naïve animals, *TNT[G]* expression in CAMEL neurons abolished this difference and impeded courtship LTM (Fig. 4C), as previously shown for olfactory aversive LTM[47]. Overall, these findings suggest that the CAMEL tool does label putative engram cells for courtship LTM.

We performed FACS followed by single cell RNA-seq on CAMEL positive putative engram neurons under three experimental conditions: wild type trained (WTT), wild type non-trained (naïve), and trained *Adyc1* mutants (Fig. 4D). A fraction of WTT animals clearly showed reduced courtship behavior compared to *wild type* naïve flies and trained *Adyc1* mutants 24 h after training (Fig. 4E). For the WTT group, we selected only trained flies with effective memory recall, i.e., with diminished courtship towards a mated female (see Fig. 4E),

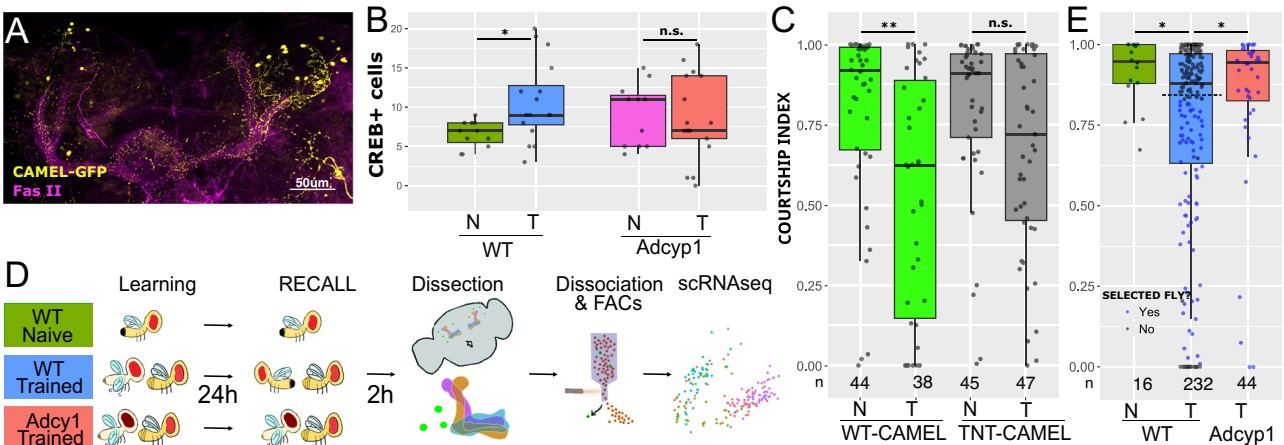

**Fig. 4 | CAMEL-GFP neurons identify the memory engram cells. A** Example of CAMEL-GFP (yellow) mushroom body co-stained with anti-Fasciclin II (magenta). Scale bar represents 50 μm. **B** Box plot showing CAMEL-GFP positive cells for naïve and trained flies of *wt* and *Adcy1* mutant backgrounds. Statistical significance was determined using a two-tailed *t*-test, *p* = 0.015 for *WT* Naïve (*WT-N*; *n* = 11) vs *WT* Trained (*WT-T*; *n* = 16), and *p* = 0.7572 for *Adcy1* Naïve (Adcy1-N; *n* = 11) vs *Adcy1* Trained (Adcy1-T; *n* = 17). Box: 25th-75th percentile, whiskers: full data range, line: median Source data are provided as a Source Data file. **C** Box plot showing court-ship indices (CIs) for *CAMEL>eGFP* (WT CAMEL) and *CAMEL>TNT^G* for LTM assays 24 h after training. Statistical significance was determined using a two-tailed Mann-Whitney test *p* = 0.004 for WT CAMEL Naïve (WT-CAMEL N) vs WT CAMEL Trained (WT-CAMEL T), *p* = 0.214 for *TNT^G* CAMEL naïve (TNT-CAMEL N) vs *TNT^G* CAMEL Trained (TNT-CAMEL T). Number of flies tested for each condition is shown under corresponding dot plot. **D** Experimental scheme to isolate CAMEL positive MB neurons. Three experimental groups of flies were used: WT naïve (green), WT trained (blue) and *Adcy1* mutant trained (red). WT and *Adcy1* mutant trained ani-mals were tested for 10 min, and 2 h later brains were dissected, dissociated, selected by FACs and sequenced (scRNAseq). **E** Box plot showing courtship indices (CIs) for the three experimental groups for LTM assays. Statistical significance was determined using a two-tailed Mann-Whitney test *p* = 0.0214 for WT Naïve vs WT Trained, *p* = 0.0459 for WT Trained vs *Adcy1* Trained, and *p* = 0.3395 for WT Naïve vs *Adcy1* Trained. Box: 25th-75th percentile, whiskers: full data range, line: median. Number of flies tested for each condition is shown under corresponding dot plot. Selected flies used for dissection and subsequent scRNAseq were shown as blue dots, with a Courtship Index below 0.72 or at least 3 unsuccessful courtship trials (threshold marked by a dotted line in the trained condition). Source data are provided as a Source Data file.

discarding males with a null courtship index (CI). *Adcy1* mutant males also faced a mated female, with no selection afterwards. Two hours after testing (i.e., after memory reactivation, in the case of WTT), we dissected and dissociated 40–60 brains per condition, removing the optic lobes. CAMEL positive cells were isolated using FACS (Fig. 4D). The percentage of GFP positive cells identified by FACS was approxi-mately double for WTT (0.24%) compared to naïve (0.12%) (Fig. S3D), as expected based on confocal imaging (compare Fig. 4B and Fig. S3). We performed deep scRNA-seq (Smartseq2) on CAMEL-positive cells from all three groups. After filtering and quality control, we obtained 48, 37 and 104 cells from brains of naïve, *Adcy1* mutant, and WTT flies, respectively.

**Identification of putative memory engram cells**
Principal Components Analysis (PCA) of the transcriptomic profiles of CREB-activated MB neurons of WTT flies revealed two distinct clusters, named WTT-1 and WTT-2 (Fig. 5A). To understand the identity of these cells, we determined the expression of known general MB identity genes, such as *ey, Dop1R2*, O*amb, mub, dac* and *prt* (Fig. 5B and Fig. S4). These markers were expressed in most WTT-1 cells, but not in WTT-2. This suggests that the WTT-1 population likely represents CREB acti-vated MB cells contributing to memory, while the WTT-2 population represents CREB activated non-MB cells. To understand the similarities of neurons from WTT flies with the other two experimental groups, we added cells from naïve (no training) and *Adcy1* mutant (ineffective training) flies to the PCA analysis (Fig. 5C). This distinguished five neuronal populations, clusters 1–5 (Fig. 5D). Visual representation of the most differentially expressed genes (DEGs) among the five clusters showed clear differentiating transcriptional signatures, although very similar between cluster 1 and 2 (Fig. 5E shows the top 10 DEGs and Fig. S5 shows the top 20 DEGs).

Cluster 1 and 2 did not express general MB markers (*ey, Dop1R2, oamb, dac, mub,* and *prt*), suggesting that these clusters represent CREB-activated non-MB cells (Fig. 5F). Regarding the cluster composition, Cluster 1–2 contained primarily neurons from naïve flies and cluster WTT-2 of trained animals (81% and 100%, respectively) (Fig. 5G). In contrast, MB neurons of the WTT-1 group were primarily seen in cluster 4 (77%) and cluster 5 (43%) (Fig. 5G), and most cells from these clusters did express general MB markers (Fig. 5F). Cluster 5 was comprised of cells from all conditions, but still the most represented cell population is WTT-1 (Fig. 5G). Cluster 3 was also comprised of cells from all conditions, but the majority (53%) belong to trained *Adcy1* mutant flies (Fig. 5G). Based on the expression of MB marker genes, such as *Oamb, mub* and *prt*, cluster 3 contained mostly MB cells (Fig. 5F). However, expression of other MB markers related to memory formation (such as *Dop1R2* and *dac*) was low. To assign cells to indi-vidual MB lobes, we examined the expression of several MB lobe marker genes[50]. Expression of the αβ and γ cell marker *sNPF* was seen in most cells of cluster 3–5 and absent in the majority of cells from cluster 1 and 2 (Fig. 5F). α'β' and γ neurons express *trio* and *mamo*, which were highly expressed in Cluster 5. Clusters 1–2 also showed high levels of *trio* and *mamo* (Fig. 5F), which was expected, since these genes are not exclusively expressed in the MB[50]. The αβ marker *prospero* is highly expressed in cells of cluster 3–4, while the γ-specific marker *ab* is primarily expressed in cluster 5 (Fig. 5F). We also examined an addi-tional set of 30 markers for different MB cell types[50], which overall suggests that cluster 1–2, 3–4, and 5 have a transcriptional profile reflecting non-MB, αβ, and γ neurons, respectively (Fig. S4).

GO enrichment analysis of the most DEGs from Cluster 1 and 2 revealed enrichment of terms related to axons, synapses, calcium ion binding, GTPase activity, mitochondria, cellular respiration, and transmembrane transporter activity (Fig. S5 and Supplementary Data 5). These genes might reflect general neuron activity that would be expected in CREB activated cells. Cluster 3, mostly composed of *Adcy1* mutant cells, was enriched for ribosomal and endoplasmic reticulum components, as well as in genes related to ubiquitin and catabolic processes (Fig. S5 and Supplementary Data 5). Cluster 4, which is mostly composed of WTT-1 cells with αβ lobe identity, was

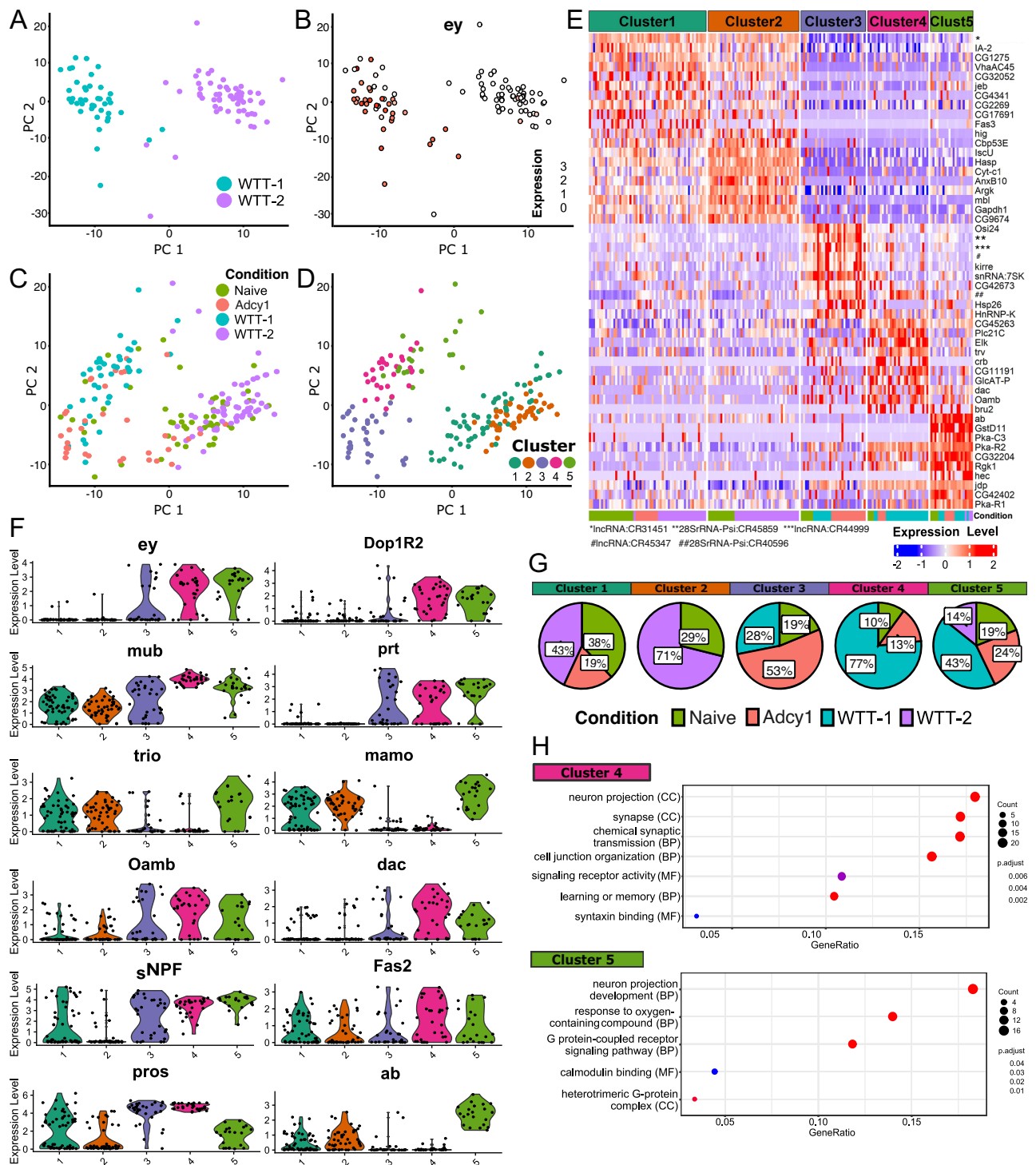

**Fig. 5 | Analysis of scRNAseq data differentiates two memory related clusters.**
**A** Principal Component Analysis (PCA) of WT trained flies with successful retrieval
($K = 25$). WTT-1 and WTT-2 are shown in light blue and purple, respectively. Each dot
represents a neuron. **B** *ey* expression (white to red scale) in the PCA of WT trained flies.
**C** PCA revealing the experimental condition for each cell: trained (WTT-1 and WTT-2),
naïve (green) and trained *Adcy1* (red). **D** The five clusters as defined by PCA ($K = 15$).
**E** Composition of the five clusters by experimental conditions (naïve, *Adcy1*, WTT-1
and 2) by percentage. **F** Violin plots for general mushroom body and lobe marker
genes for the five clusters. **G** Heatmap of the 10 most DEGs among the five clusters.
The lower bar in the graph indicates the experimental condition for each sequenced
neuron. **H** Most representative GO terms (FDR < 0.05) for the 4 and 5 clusters, indi-
cating cell component (CC), biological processes (BP) and molecular function (MF).

enriched for the GO term 'learning and memory' as well as terms
related to synaptic transmission (Fig. 5H and Supplementary Data 5).
Cluster 5, which is enriched for WTT-1 cells from MB γ neurons, showed
enrichment of terms related to memory signaling pathways, including
"G protein-coupled receptor signaling" and "calmodulin binding". Both

clusters 4 and 5 are therefore transcriptionally enriched in processes
classically associated with LTM formation (Fig. 5H and Supplemen-
tary Data 5).

We also looked at the expression of the 68 core TIGs that we
identified as part of the transcriptional trace of memory consolidation

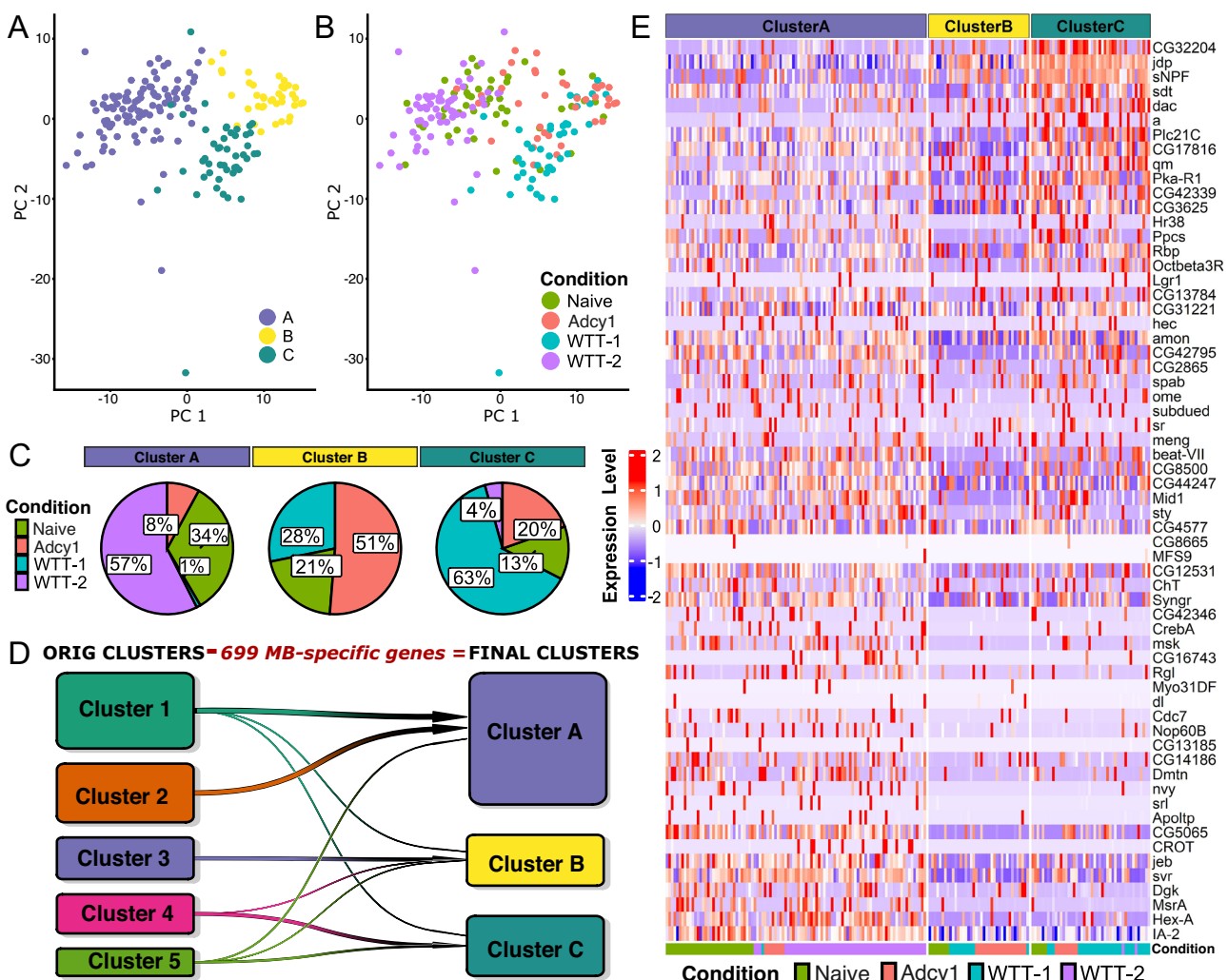

**Fig. 6 | Removal of MB- and lobe-specific genes still identifies the putative memory-forming neuronal population.** **A** The three clusters as defined by PCA (*K* = 25). Cluster A, B and C are marked in blue, yellow and green, respectively. **B** PCA revealing the experimental condition for each cell: WTT-1 (light blue), WTT-2 (purple), naïve (green) and *Adcy1* trained (red) conditions. **C** Percentage of cells belonging to each of the three clusters by experimental conditions. **D** Transition plot representing the equivalence between clusters 1–5 and clusters ABC. **E** Heatmap of the 68 core up-regulated TIGs after learning obtained by bulk RNAseq following courtship conditioning. Genes are ordered by their expression levels in cluster C.

at 1 h after courtship LTM training (Fig. 3). Cluster 3, which is composed mostly of MB cells from the memory deficient *Adcy1* mutant, showed low expression of the 68 TIGs. Non-MB clusters 1–2 displayed strong induction consistent with an activated-CREB response. Clusters 4–5, representing MB cells from trained animals, exhibited renewed expression indicative of maintenance in expression of learning and memory consolidation genes (Fig. S5).

### Identification of cell-type-independent memory genes

Our data suggests that the formation of Clusters 1–5 was driven by a combination of cell-type specific markers as well as components of the CREB-activated memory transcriptome that were not cell-type specific. To identify memory candidate genes independent of MB identity we removed 699 genes that showed enriched expression in the MB in our INTACT data (Supplementary Data 6 and Fig. S1) and other studies[50,51]. PCA analysis of the remaining genes revealed three clusters, A, B, and C (Fig. 6A). Cluster A was mainly formed by non-MB neurons from naïve and WTT-2, with 90% of cells derived from the previous Clusters 1 and 2 (Fig. 6B–D). Cluster B was mostly composed of αβ neurons from the *Adcy1* mutant (51%), with most cells derived from the previous cluster 3 (Fig. 6B–D). Cluster C was mostly composed of WTT-1 cells (63%), thus

merging previous cluster 4 (αβ neurons) and 5 (γ neurons) (Fig. 6B–D). Again, all clusters contain some naïve and *Adcy1* mutant cells.

Intriguingly, the GO terms enriched among the top DEGs of Clusters A and B were very similar, including enzyme binding, synapse organization, proteolysis, and cytoplasmic vesicle, in spite of the different neuronal identity (Fig. S6 and Supplementary Data 7). Cluster C showed enrichment primarily in genes related to energy metabolism, translation, neuron projection development, and chemical synaptic transmission, which were also all enriched terms identified clusters A and B (Fig. S6 and Supplementary Data 7). Despite common GO enrichment, the genes driving the enrichment are largely different between the three clusters, suggesting the possibility of alternate functional output (Fig. S6). Interestingly, processes such as energy metabolism, translation, and synaptic transmission were also identified to be enriched in the MB specific transcriptional trace of memory formation identified by INTACT (Fig. 1). Consistently, we observed high expression of the 68 core TIGs from memory consolidation in clusters A and C, but not B (Fig. 6E). To summarize, the new cluster C was mainly composed by MB αβ and γ WTT-1 cells of previous cluster 4 and 5, revealing candidate genes potentially relevant in memory processes, independent of MB cell identity.

## Loss-of-function screen of candidate genes

As a proof of causality, we compiled a list of candidate genes for functional testing. We first identified the 30 most DEGs among Clusters A, B and C (Fig. S7A). Among these, we selected genes that; (i) showed high expression levels in most of the individual neurons within each cluster, (ii) had a clear mammalian ortholog (at least a score of 4 via DIOPT v9.1 from Flybase), and (iii) had a *UAS-RNAi* line available from the Bloomington Stock *Drosophila* Center (Fig. S7 and Supplementary Data 8). Genes with known roles in memory (such as *pkc53E* and *Pde4*) or RNAi processing (such as *AGO1*), were discarded. We compiled a list of the highest expressed Cluster C genes that showed high expression in at least 50% of cells and selected additional candidate genes from this group (Fig. S7 and Supplementary Data 8). Since cells from all conditions (WTT, naïve, and *Adcy1* mutants) are present in cluster C, we also selected candidate genes that were highly expressed in more than 80% of WTT-1 neurons and in less than 20% of neurons from the other conditions, favoring uncharacterized genes (annotated with CG numbers) (Fig. S7 and Supplementary Data 8). Based on these criteria, we selected 7 genes from cluster A, 6 genes from cluster B, 18 genes from cluster C, and 10 genes from cluster C>80% WTT1. Finally, we selected 7 genes that were consistently induced across different studies using INTACT-RNAseq (Fig. 3C), for a total of 48 candidate genes (Fig. 7A).

To test the potential involvement of these candidate genes in courtship LTM, we performed a memory screen using *UAS-RNAi* lines under the control of specific MB-Gal4 drivers expressed in αβ and γ lobes; either RH1406-Gal4, or MB247-Gal4 (Figs. S1 and S9). We combined them with either *UAS-dcr2*, to enhance RNAi effectiveness in the case of long hairpins vectors, or *tub-Gal80ts*, to avoid developmental effects (for specific details, see "Methods", Fig. S8 and Supplementary Data 9). Long hairpin RNAi transgenes were not combined with *UAS-dcr2* for *tub-Gal80ts* experiments, to avoid potentially confounding effects of high levels of *dcr2* expression at 29 °C. The positive hits from the screen are shown in Fig. 7B–E and the full set of experiments can be seen in Fig. S8. Control flies displayed normal LTM, with a significant decrease in courtship behavior in trained animals compared to naïve (Fig. 7B–E and Fig. S8). RNAi knockdown of 19 candidate genes abolished LTM, with no significant difference observed in courtship behavior between trained and naïve flies (Fig. 7C–E). To test if any memory defects were due to defective development of the MB, we stained the knocked-down MBs for each positive gene with TRIO and FasII antibodies, which label α'β'/γ and αβ/γ lobes, respectively. For three genes, *LpR1, Smox*, and *Teneurin-A*, we saw clear defects in morphology and therefore excluded them as candidate memory genes (Fig S9). The 16 positive hits without major structural alterations were classified according to their functions (Fig. 7F), which included signaling (MAPK-*CG7378*, WNT-*pan*), transcription (*Hr38, sr, cpo, fs(1)h*), ubiquitination (*CG2915, CG17691, CG11700*), and synapses (*sur* and *coracle*). Among the tested genes, 50% from cluster C, 30% of cluster C>80% WTT-1, and 43% of tested core TIGs from INTACT RNAseq were confirmed as memory regulators (Fig. 7G). In contrast, only 14% of cluster A and 17% of cluster B genes were confirmed (Fig. 7G). Our RNAi approach may include false negative results due to insufficient knockdown, or false positives due to off-target effects. Nonetheless, the overrepresentation of positive genes from cluster C and from core TIGs, supports the idea that our transcriptome analyses have revealed novel memory genes participating in memory formation, storage, and possibly recall.

## Identification of Drosophila memory ARGs

Of the positive hits from our memory screen, two transcription factors, *Hr38* and *stripe* (*sr*), were previously identified as neuron activity induced genes[11,52] and therefore represent candidate ARGs that may govern the transcriptional response to courtship LTM. When *Hr38* and *sr* RNAi were induced in the MB 1 day before training using Gal80ts, we

observed defects only in LTM and not in STM compared to genetic background controls containing Gal4 and Gal80ts alone (Fig. 7B–D). In contrast, when we performed knockdown starting 2 days prior to training we did observe STM defects for Hr38, but not Sr (Fig. S8I). This indicates that in the adult MB, Sr may have a very specific role in LTM. Hr38, on the other hand, seems to have a broader role, since a 2-day knockdown yielded STM and LTM defects. Two-day *Hr38* knockdown in the MB also caused reduced naïve courtship (Fig. S8I), consistent with a broader role for Hr38 in memory and social behavior[52]. Flies that were heterozygous for the Hr38 and Sr *UAS-RNAi* transgenes, with no Gal4 or Gal80, showed normal memory, demonstrating that defects are not induced by the temperature shift alone (Fig. S8J). While off target effects are possible with RNAi, the lines used here have no homology to other genes, and have shown no phenotype when expressed with several other tissue specific Gal4 drivers[53–60], indicating a low probability of off target effects.

Memory ARGs are not well described for any *Drosophila* memory paradigm. We therefore compiled a list of known neuron activity-induced genes in *Drosophila* ($n = 14$)[11], as well as fly orthologs of well characterized human neuron activity-induced genes, including *Arc1* (human *ARC*), *Jra* (human *JUN*), *kayak* (human *FOS*), and *Dysf* (human *NPAS4*). These 18 genes were examined for differential induction across the MB memory transcriptome time course and in our single cell clusters (Fig. 8A, B and S10). Hr38 and Sr stood out in our memory time course analysis compared to the other candidate ARGs because they were strongly and immediately induced at a much higher level in the MB than in WH (Fig. 8A, B). Indeed, *Hr38* and *sr* are among the strongest induced transcripts of the MB-specific TIGs (Supplementary Data 2). Transcript levels of *Hr38* and *sr* decline after 1hDT but remain significantly higher than in naïve MBs until 1hAT (Fig. 8A, B). We validated Hr38 and sr expression at 1hAT using MB INTACT followed by qPCR. *Hr38* and *sr* were clearly induced by LTM training (Fig. 8C), but not by STM training (Fig. 8D), further supporting a role in LTM for these TFs. *Hr38* was also clearly induced in the memory recall cluster 5 containing WTT-1 neurons (Fig. S5). In our cell-type-independent ABC clusters, the 16 candidate ARGs were most induced in cluster A, which is composed of non-MB CREB activated cells. However, a number of Cluster C cells also showed induction of some ARGs, including *Hr38* and *sr* (Fig. S10). Overall, these data suggest that the transcription factors *Hr38* and *sr*, known ARGs, may help to shape the transcriptional trace of courtship LTM.

## Hr38 and Sr binding to TIGs and engram genes

To explore the potential mechanisms underlying courtship LTM training-induced gene expression, we analyzed chromatin structure using Assay for Transposase Accessible Chromatin followed by sequencing (ATAC-seq) on INTACT-isolated MB nuclei. Interestingly, both *Hr38* and *sr* show very high accessibility near, or directly surrounding their transcriptional start sites, much higher than the accessibility of even the most highly expressed MB genes (Fig. S11). The highly accessible chromatin landscape of *Hr38* and *sr* promoters in the MB could facilitate their immediate and robust induction following courtship LTM training. The CrebB transcription factor has been suggested to induce memory ARGs in other systems. Interestingly, we found that CrebB binding sites, obtained using publicly available whole embryo ChIP-seq data from the ENCODE data portal[61], directly overlapped with MB-specific ATAC-seq peaks in the TSSs of *Hr38* and *sr* (Fig. 8E). The presence of CrebB binding at these MB accessible sites suggests that *Hr38* and *sr* could be direct targets of CrebB.

We next investigated whether the MB-specific transcriptional trace of courtship LTM could be mediated by Hr38, Sr, and/or CrebB. We identified Hr38, Sr and CrebB binding sites using publicly available whole organism (embryo or prepupa) ChIP-seq data[61]. Potential MB binding sites for Hr38, Sr and CrebB were identified by overlapping ChIP binding sites with MB-specific ATAC-seq peaks, which were highly

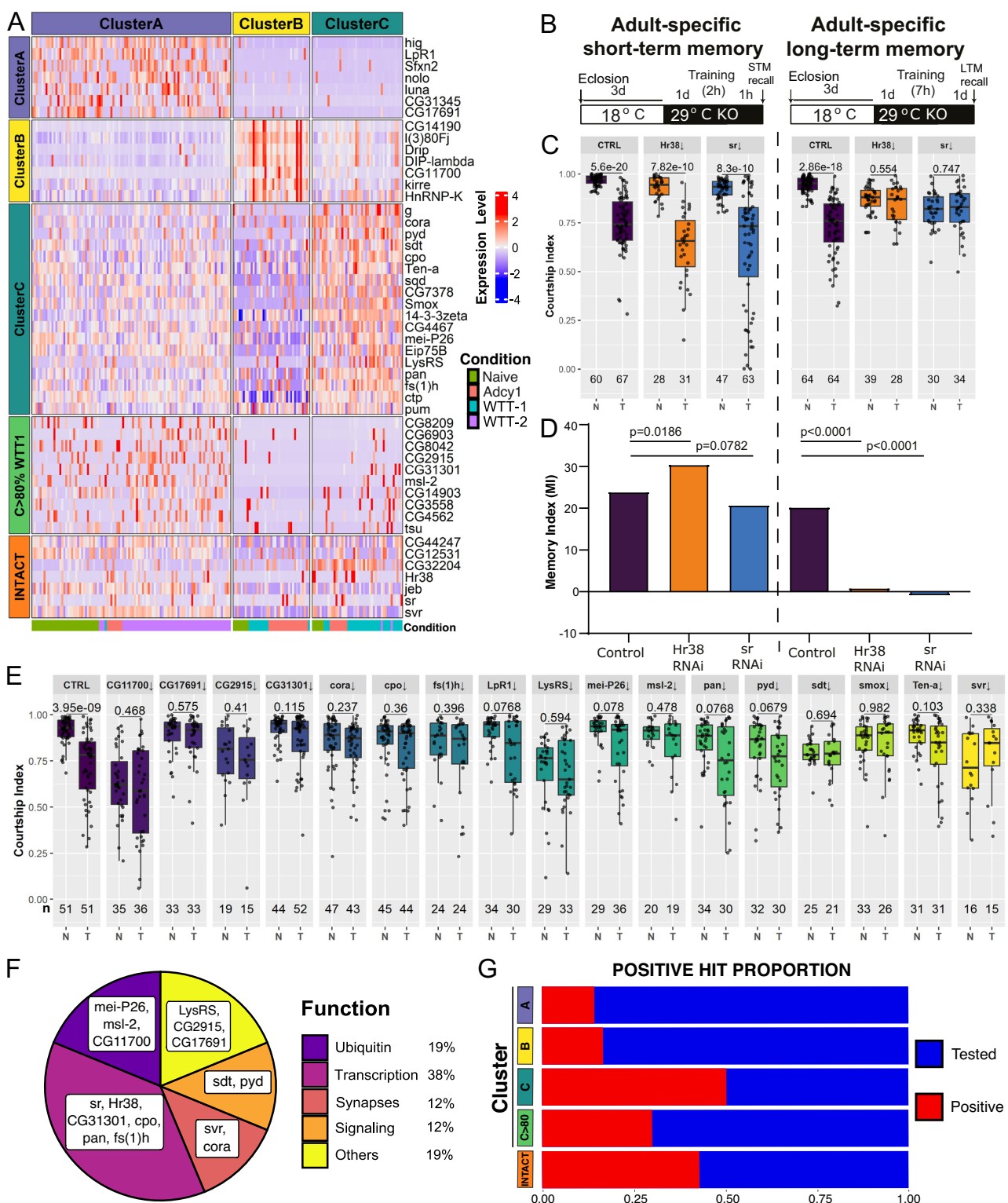

consistent between two biological replicates (Fig. S11 and Supplementary Data 10). If experimentally identified ChIP binding sites from embryo or pupae are present at regions of MB open chromatin, it is very likely that these represent MB binding sites. It is possible that the MB has extra binding sites that are not detected in embryo, and these would be missed in our analysis. Therefore, our data may include false negatives, but our approach also has a strong advantage that it is not likely to produce false positives. Of the 756 MB TIGs that form the MB-specific courtship LTM trace, 306 were bound by either CrebB, Hr38, or Sr and more than half of these were bound by two or more of these

TFs (Fig. 8F and Supplementary Data 11). This suggests a high level of redundancy in the activation of TIGs. Interestingly, the 306 TF-bound TIGs show late and post training activation that is significantly greater in the MB than in the WH (Fig. 8G). This MB-specific memory trace is observed when looking at all 306 TF bound TIGs and those bound by two or more TFs ($n = 163$), CrebB only ($n = 74$), or Sr only ($n = 63$) (Fig. 8G). In addition, we examined TF binding sites among the top 20 differentiating genes in Clusters A, B, and C, from our cell-type-independent scRNAseq analysis. Interestingly, Cluster C genes showed a higher frequency of TF binding, with the majority bound by Sr and

**Fig. 7 | A functional screen identifies new genes governing courtship LTM.**
**A** Heatmap of the 48 candidate genes to be tested by courtship conditioning, from cluster A, B, C, C>80% WTT1 and INTACT RNAseq. **B** Schematic of approach for achieving adult-specific knockdown in the MB of *Hr38* and *sr*. AttP2 insertions from the TRiP collection and the genetic background control strain BDSC36303, were crossed to *tub-gal80ts; R14H06-gal4* and the progeny were raised at 18 °C and transferred to 29 °C 24 h before STM and LTM training. **C** Box plot showing courtship indices (CIs) for naïve (N) and trained (T) flies with MB-specific Hr38-RNAi, sr-RNAi, and the respective genetic background controls. Statistical significance between naïve and trained flies was determined using a two-tailed Mann-Whitney test. Box: 25th-75th percentile, whiskers: full data range, line: median. Number of flies tested for each condition is shown under corresponding box

plot. **D** Bar graph showing corresponding memory index (MI) derived from the CI (see "Methods") Statistical significance between MIs was determined using a two-tailed randomization test with 10,000 replicates. **E** Box plots showing courtship indices (CIs) for naïve (N) and trained (T) flies for candidate genes whose downregulation in the MB abolishes LTM. A control example is shown. Statistical significance between naïve and trained flies was determined using a two-tailed Mann-Whitney test. Box: 25th-75th percentile, whiskers: full data range, line: median. Number of flies tested for each condition is shown under corresponding box plot. Source data are provided as a Source Data file. **F** Classification of positive hits according to function. **G** Percentage of positive hits versus tested candidate genes from each category of cluster (A, B, C and C>80% WTT1) as well as from INTACT RNAseq data.

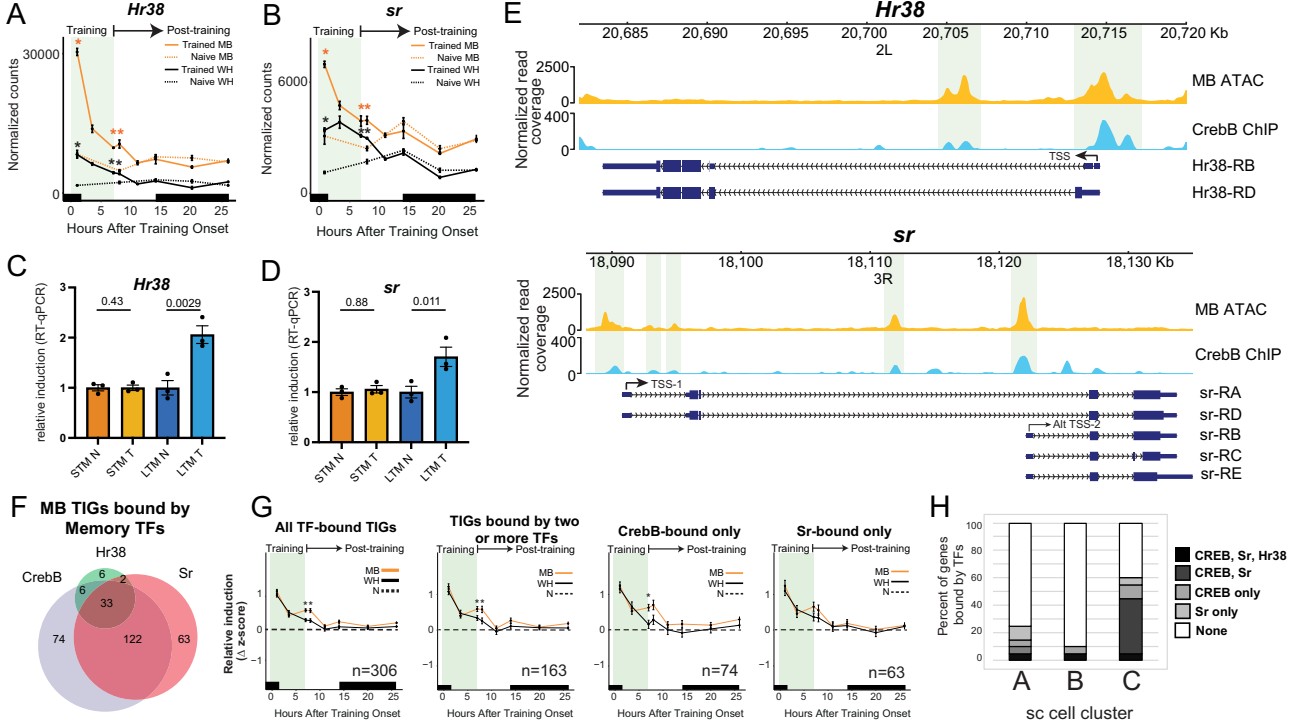

**Fig. 8 | *Hr38* and *sr* bind MB-specific TIGs and are potential CrebB targets.**
Normalized transcript levels of **A** *Hr38* and **B** *sr*. Significance differences identified from differential expression analysis between trained and time-of-day matched naïve flies are indicated ($N = 3$, Wald's test, *FDR < 0.1). Black bars indicate night-time. Real-time quantitative PCR analysis of **C** *Hr38* and **D** *sr* induction after STM and LTM training. Statistical significance was determined using a two-tailed *t*-test, $N = 3$, *p* values are indicated. **E** *Hr38* and *sr* genomic tracks displaying MB chromatin accessibility (ATAC-seq−top panel), CrebB binding signal (ChIP-seq−middle panel), and transcript isoforms of *Hr38* and *sr* (bottom panel). Regions that overlap

between CrebB binding signal and accessible MB chromatin are highlighted in light green. **F** Overlap of binding sites in regions of accessible MB chromatin for the transcription factors (TFs) Hr38, CrebB and Sr with training induced genes (TIGs) identified in the MB during LTM formation. **G** Relative z-score induction during LTM formation for all TF-bound MB TIGs, MB TIGs bound by two or more TFs, CrebB-bound only, or Sr-bound only. *P* values were determined using a two tailed *t*-test (*n* values are indicated, *p < 0.05). Black bars indicate nighttime. **H** Percentage of the top 20 genes from the clusters A, B and C that were bound by Hr38, CrebB, and/or Sr. All error bars indicate the standard error of the mean.

CrebB (55%), compared to Cluster A (25%) and Cluster B (10%) (Fig. 8H). Overall, these findings suggest that CrebB, Hr38, and Sr may cooperate to activate expression of a MB-specific transcriptional trace of courtship LTM training that persists from early training, into memory recall.

**Overlap with other transcriptome datasets**
Several studies have been performed in *Drosophila* that analyze the transcriptome of different neuronal tissues in response to memory and social interaction[25–31,44,62]. We found a significant overlap of genes that were induced in two previous datasets produced in our laboratory compared to this study (Fig. 3), however, we wanted to compare more broadly. Winbush et al., looked at gene expression changes in the WH, 24 h after courtship LTM training[26]. Interestingly, a transcript specific analysis identified about 500 induced genes, which showed significant overlap with MB and WH TIGs identified here, including important memory genes such as *svr* and *PKA-R1* (Supplementary Data 12).

Another study by Agrawal et al. compared gene expression changes in dopaminergic neurons (DANs) in group housed (GH) flies compared to solitary housed (SH)[62]. This social behavior paradigm partially mimics courtship conditioning since naïve flies which are housed alone are compared to trained flies that experienced a social interaction. Interestingly, we found many WH and MB TIGs that were differentially expressed in DANs (Supplementary Data 12). This included the ARGs *Hr38*, *sr*, and *CrebA*.

Other *Drosophila* memory transcriptome studies have focused on completely unrelated behavioral paradigms including aversive olfactory conditioning. Crocker et al. analyzed transcriptome changes occurring after olfactory memory training in several different neuronal subsets isolated by patch clamp pipetting[29]. While the total overlap of genes was low with this study, we did find several MB and WH TIGs overlapping with memory induced genes here including *Hr38*, *sr*, *svr*, and *sNPF* (Supplementary Data 12). Widmer et al. used targeted DamID

to profile RNA polymerase II occupancy as a proxy for gene expression over 4 different time intervals spanning up to 72 h after training[28]. Again, several induced genes were also found among our MB and WH TIGs. Therefore, despite the use of memory assays that are very different from courtship conditioning, we still observe induction of some key memory genes across different studies.

We also looked at the expression of memory-induced genes from different studies in our single cell clusters. Many of these genes showed high expression levels in Cluster C, representing potential MB engram cells, but were especially enriched in Cluster A, representing non-MB CREB activated cells (Figs. S12 and S13). In contrast, activation was generally not as high in cluster B, mostly representative of *Adcy1* mutant cells.

## Discussion

In this study we performed a transcriptome time course analysis in memory neurons of the *Drosophila* MB, during courtship conditioning. This analysis revealed a transcriptional trace of memory consolidation that is active in the MB, and not the WH, near the end of the 7 h training period and after training, during the time when LTM consolidation is occurring. The memory trace is comprised of genes that contribute to known cellular processes underlying LTM, including actin cytoskeleton remodeling, energy metabolism, translation, and cAMP signaling. In addition, single cell RNAseq analysis identified a select population of CREB activated MB cells that showed a persistent transcriptional signature of memory formation. We showed that this neuronal population is required for courtship LTM and shares some common genes with the MB-specific transcriptional trace of early learning and memory consolidation. In a proof-of-causality screen we identified 16 genes required for memory. Among these genes, we identified two candidate memory ARGs for *Drosophila*, *Hr38* and *sr*. These genes are orthologs of mammalian IEGs, *NR4A2* and *EGR1*, which have been described to be important for learning and memory[63,64] *Hr38* and *sr* have previously been characterized as ARGs induced during social interactions[52,62], ethanol exposure[65], and general induced neuronal activity[11], but have not been characterized in the context of memory. Overall, this work significantly advances our knowledge of the transcriptional programs that are induced to facilitate LTM and identifies critical TFs that may control these programs.

In mammals, it is well established that neuronal activity induces a CREB-dependent transcriptional wave that, in turn, induces thousands of ARGs[66]. Among them are IEGs that are transiently expressed during LTM formation and are widely used to define cellular engrams[12,46]. In recent years there is evidence that IEGs vary depending on patterns of neuronal activity, stimuli, and cell type[67,68]. It has been proposed that information is sorted into different engrams that are determined by distinct IEGs, as demonstrated in mouse fear conditioning for two well-known IEGs[63,69]. This resembles what happens in *Drosophila*: the *c-fos* homolog *Kayak* was up-regulated in olfactory memory[70], whereas in our study using courtship conditioning we only detected *Hr38* and *sr*. Many other known ARGs did not show up-regulation under our conditions, supporting the idea of specificity in ARGs and IEGs underlying different types of memory.

We compared our memory transcriptome dataset from courtship LTM training with previous studies performed in *Drosophila* that analyzed the transcriptome in response to social interaction and memory[25–31,44,62]. In the context of social interaction, Agrawal et al. identified expression changes in DANs for *Hr38 and sr* in GH flies compared to isolated flies[62]. They used targeted *Hr38* and *sr* RNAi to demonstrate that these ARGs were required in DANs for differential sleep behavior observed when flies are GH. Recently, social enrichment has been linked to memory formation[48], suggesting an explanation for the overlap between genes induced by socialization, and courtship LTM training.

Interestingly, the expression of memory induced genes in our scRNAseq data was strongest with Winbush et al., a study that looked at memory induced genes 24 h after courtship conditioning[26], very similar to the conditions from which we isolated CAMEL positive neurons. Winbush et al. identified many of the top DEGs from cluster 4–5 and C, as well as positive validated hits *pan* and *cpo*. Comparison with memory-transcriptome studies that used different memory paradigms rendered some common genes, such as *Hr38, svr* and *sr*[28,29]. However, despite the observed overlap between memory-induced genes identified here and in other studies, most genes are unique to a single dataset. These differences may be explained by several factors including the use of different memory paradigms, different tissues, and different gene expression analysis techniques. But even when conditions are similar, several factors may contribute to variability. First, all studies that we compared were done in different genetic backgrounds, and it is known that gene expression is sensitive to genetic background. Second, statistical power may vary between datasets for different genes leading to false negatives and positives. The number of biological replicates used and the variability between the replicates will affect the statistical significance of gene induction. Some genes may narrowly miss a threshold cutoff in one dataset. Indeed, we found that many genes that did not meet a statistical threshold for induction in all datasets, did show a trend toward induction in all (Fig. 3B). Third, it is very likely that memory gene induction is in part stochastic in nature and dependent on the current internal state of each individual fly. For example, some flies may be predisposed to better memory formation and may already have higher expression of critical genes. Indeed, individuality does exist in *Drosophila* and is at least in part mediated by stochastic neurodevelopmental processes[71–73].

One of the goals of our scRNAseq experiment was to identify a transcriptional trace of the memory engram. Due to the very low number of CAMEL positive neurons unveiled by confocal imaging, we sampled relatively low numbers of cells. Low sampling might miss rare engram neuronal subtypes, such as those related to memory reactivation. In mammals, the number of reactivated neurons after recall is about 10%[46], suggesting that their number might be scarce also in *Drosophila*. However, despite low availability of cells, dissociation and sorting allowed us to obtain enough CAMEL positive cells from several individuals to perform robust clustering analysis which permitted to distinguish CREB activated MB neurons from non-MB cells and revealed one cluster that was enriched for cells from *Adcy1* mutant animals who do not form LTM. After removal of MB enriched genes, we still identified 3 clusters, with cluster A containing non-MB neurons, cluster B containing mostly *Adcy1* mutant cells and cluster C containing MB neurons mostly from trained animals. Memory engram cells are most likely contained within cluster C, however, we did not identify a specific cluster containing only cells from trained animals. All three experimental conditions were present in each cluster, although at different proportions. We functionally tested some candidate genes that were up-regulated mainly in WTT-1 and not in control conditions (Fig. S7), but did not increase the number of positive hits under this approach (Fig. 7G). Taken together, this suggests that we could not fully distinguish memory-reactivated engram neurons from neurons activated from different experiences (i.e., memory formation or learning unrelated to courtship LTM training). As stated above, it might be that memory reactivated cells are too rare to be identified under this experimental approach. Increasing the number of sequenced neurons might uncover the engram cells. However, the presence of all three conditions (trained, naïve and A*dcy1* mutant) in Cluster C, together with the functional analysis of genes enriched in WTT-1, suggests an alternative possibility: established engram neurons might share a similar transcriptional profile to neurons that have activated CREB during early learning or consolidation. Indeed, we find

many similarities in the genes induced during early learning and social interactions, compared to consolidation and after recall.

## Methods

### Fly stocks and genetics

*Drosophila melanogaster* stocks were reared on a standard medium (cornmeal-sucrose-agar), supplemented with the mold inhibitors methyl paraben and propanoic acid, at 25 °C in 70% humidity with a 12h:12h light/dark cycle. Wild-type female flies used in this study were an in-house generated Canton-S/Oregon-R mixed genetic background called Nijmegen wild type. *5xUAS-Unc84::2xGFP* flies were a gift from G.L. Henry[41]. RNAi lines stocks were obtained from the Bloomington *Drosophila* Stock Center (BDSC; Bloomington, USA), or the Vienna Drosophila Resource Center (VDRC; Vienna, Austria). *R14H06-GAL4* flies express GAL4 under the control of a MB specific enhancer for *Adcy1* (BDSC #48677)[42,74]. The CAMEL reporter tool is the result of combining *6xCRE-splitGal4*^AD and *UAS-eGFP;R21B06-splitGal4*^DBD stocks, that were kindly donated by Dr Jan Pielage[47]. *Adcy1^2080^; 6xCRE-splitGal4^AD and TNT^G;6xCRE-splitGal4^AD* stocks were made in our laboratory and they are available upon request.

We used an isogenic heterozygous breeding strategy to produce experimental and genetic background control flies. Briefly, a common driver line containing *R14H06-Gal4* or *MB247-Gal4*, as well as accessory transgenes such as UAS-dcr2 and temperature sensitive GAL80 (GAL80^ts)[75] (when relevant), was crossed to RNAi strains, and their isogenic controls (Supplementary Data 9). UAS-RNAi stocks were generated by the Transgenic RNAi project (TRiP, Harvard University)[76] or as part of the VDRC KK library (Supplementary Data 9). Different genetic background control lines were used depending on the RNAi line (Supplementary Data 9). Temporal control of GAL4 was achieved with GAL80^ts, which is expressed ubiquitously under control of the $\alpha$Tub84B promoter. For Gal80^ts experiments, flies were raised at 18 °C, which restricts GAL4 activity, preventing RNAi knockdown during development. Knockdown was then initiated 1 or 2 days prior to courtship conditioning by moving flies to 29 °C. All experimental crosses, including isogenic parental and test genotypes are shown in Supplementary Data 9.

### Courtship conditioning assay

Courtship conditioning was performed as previously described with minor modifications[39]. Newly eclosed F1 male flies were collected and isolated for 5 days in individual wells of a 96-well block containing 500 μL of media. F1 male flies were split into two cohorts—trained and naïve. For STM, male flies were trained by pairing with an unreceptive, recently-mated female fly for 2 h and then placed back into isolation for 1 h—referred to as the rest period. For LTM, male flies were trained using a single 7-h training session, followed by re-isolation and a rest period of 20–24 h. Following the rest period, courtship activity was measured for each individual naïve and trained male fly by pairing with a new mated female. For every male-female fly pair, a CI was determined by calculating the percentage of time spent on courtship within a 10-min period. The memory index (MI), which represents the percentage reduction in courtship behavior between trained and naïve flies, was calculated using the formula: $MI = (CI_{naïve} - CI_{trained})/CI_{naïve}$[77]. Statistical analysis between naïve and trained CIs was performed using a two-tailed Mann-Whitney test, with outliers removed by GraphPad Prism (v9.5.1) using the ROUT method with the false discovery rate set at the default value of 1%. Statistical comparison between the MIs of knockdown and control genotypes was performed using a randomization test with 10,000 bootstrap replicates[39].

### Memory transcriptome time course sample collections

Male flies homozygous for *UAS-Unc84::2xGFP; R14H06-GAL4* were crossed to Nijmegen wild type virgin females. F1 heterozygote males were socially isolated for 5 days, followed by pairing with an unreceptive female for courtship LTM training. Male flies were then collected at various time-points during LTM formation by flash freezing in liquid nitrogen. Specifically, trained males were collected at three time-points during the courtship training period (DT; 1 h, 3.5 h, 7 h) and at five time-points after training (AT; 1 h, 4 h, 7 h, 13 h, 19 h). Naïve male flies were also collected at five time-points (corresponding to 1hDT, 7hDT/1hAT, 7hAT, 13hAT, 19hAT) to act as time-of-day controls. For most datapoints, three individual biological replicates containing ~50 heads were collected, with no less than two for any one timepoint.

### Isolation of nuclei tagged in a specific cell-type (INTACT)

To isolate MB nuclei for downstream transcriptome and chromatin accessibility analyses, INTACT was performed as previously described[27,41]. Samples containing ~50 fly heads expressing *UAS-Unc84::GFP* under the control of *R14H06-GAL4* were ground with a pestle and homogenized in buffer containing 0.3% NP40 using a Dounce homogenizer. The nuclear extract was then filtered through a 40 μm strainer. A portion of this sample was collected to represent whole-head (WH) nuclei for RNA-sequencing. MB nuclei were immunoprecipitated from the remaining sample using an anti-GFP antibody (Invitrogen: G10362) bound to magnetic beads (Invitrogen: 10004D), according to the manufacturer's instructions. MB and WH nuclei were then processed for either RNA-seq or ATAC-seq.

### RNA-sequencing and data analysis

Total RNA was isolated from the WH nuclear fraction and immunoprecipitated MB nuclei using the Arcturus PicoPure RNA isolation kit (ThermoFisher Scientific: KIT0204) with DNase digestion performed using the RNase-free DNase kit (Qiagen: 79254) according to the manufacturer's instructions. RNA quality was assessed by visual examination of rRNA-peak integrity using the Bioanalyzer 2100 Pico RNA kit (Agilent: 5067-1513). RNA-seq libraries were prepared using the Tecan Universal Plus Total RNA-seq library preparation kit according to manufacturer's instructions. Library size and quality was assessed with the Bioanalyzer 2100 DNA high-sensitivity kit (Agilent: 5067–4626). Sequencing was performed with the Illumina NovaSeq 6000 at Genome Quebec with the S4 v1.5 200 cycle kit; read length was 100 bp for paired-end reads.

An average of 40,543,996 reads were generated across all MB ($n = 38$) and WH ($n = 38$) RNA-seq libraries generated. RNA-seq reads were processed on Compute Canada servers (StdEnv/2020). First, raw reads were lightly trimmed, and adapters clipped using Trimmomatic (v0.39)[78]. The read quality was assessed using FastQC (v0.11.9) and trimmed reads were aligned to the *Drosophila melanogaster* genome (Ensembl release 103, dm6) using STAR (v2.7.5a)[79,80]. Uniquely aligned reads with a maximum of four mismatches were counted to genes using featureCounts[81]. An average of 25,727,025 reads across all samples aligned to genes. Counts were then filtered for rRNA, non-coding RNA, genes mapped to the Y-chromosome or mitochondrial genome, and genes that had less than 150 normalized counts in 50 of the 76 sequenced MB and WH samples. After filtering, 6965 MB expressed genes were used for downstream differential expression analysis.

Differential expression analysis was done using DESeq2 (v1.30.1)[82] in Rstudio (v4.0.3). To identify genes altered by memory training, differential expression analysis was performed between trained flies and time-of-day matched naïve controls, using a cutoff of $FDR < 0.1$. To identify genes with enriched expression in the WH or MB, differential expression was performed between all MB ($n = 38$) and WH ($n = 38$) samples. With the large $n$ values in this comparison, we used more stringent cutoffs ($FDR < 0.05$ and $\log_2$ fold change $> 0.5$) to define MB enriched genes. Further analysis of data, including gene annotation, gene ontology (GO), statistical comparison between groups of genes, was performed using the R package BinfTools (https://github.com/kevincjnixon/BinfTools). Specific commands used included:

count_plot, getSym, barGene, zheat, GO_GEM and customGMT. GO analysis was performed using a custom background of 6956 expressed genes in our samples, and FDR < 0.05. Data was further visualized using the R packages ggplot2 (v3.4.2) and pheatmap (v1.0.12). Venn diagrams were created using BioVenn[83].

## ATAC-sequencing and data analysis

ATAC-seq was performed as previously described[84], with modifications for INTACT-isolated nuclei. MB nuclei were isolated from two independent replicates of ~50 naïve male flies at a time-point corresponding to 1h after memory training onset. To generate ATAC-seq libraries, bead-bound MB nuclei were suspended in 50 μL of transposase reaction mix (Tn5 Transposase, Illumina), and incubated for 30 min at 37 °C in a thermal cycler. DNA was then purified and eluted using a Qiagen MinElute Kit according to the manufacturer's instructions. Purified DNA was mixed with custom Nextera primers and High-Fidelity PCR Mastermix (NEB) and amplified. Amplified libraries were purified using a Qiagen PCR purification kit. Sequencing was performed with the Illumina NovaSeq 6000 at Genome Quebec with the S4 v1.5 200 cycle kit; read length was 100 bp for paired-end reads.

ATAC-seq reads were trimmed, and adapters clipped using Trimmomatic (v0.39). Trimmed reads were aligned to the *Drosophila melanogaster* reference genome (Ensembl release 103, dm6) using bowtie2 (v2.4.1) with the settings -X 2000 and -very-sensitive. Reads were shifted, +4 bp for the forward strand and -5 bp for the negative strand, to account for the 9-bp duplication created by the DNA repair nick of the Tn5 transpose[85]. Reads aligning to multiple loci, the mitochondrial genome, and scaffolds were filtered using samtools view (v1.11)[86]. Duplicate reads resulting from PCR amplification were identified using samtools fixmate and removed using samtools markdup, leaving 38,388,868 and 42,678,219 high-quality reads for downstream analysis. Peaks were called using MACS2 software (v2.1.2) using the settings -q 0.01 -min-length 50 and -max-gap 100[87]. Peak calling resulted in 15842 peaks identified uniquely between both samples, with 11705 consensus peaks, which were highly consistent between the two biological replicates (Fig. S11), and predominantly located near transcriptional start sites (TSSs), as expected (Fig. S11). Consensus peaks were annotated to 7488 genes using the R Package ChIPseeker (v1.26.2)[88]. DiffBind (v3.0.15) was used to determine the fraction of reads in peaks calculated (FRiP). The two replicates had a FRiP of >0.3 for inclusion, as per ENCODE standards[89].

For visualization of ATAC data, promoter and genomic regions were extracted using the R annotation package TxDb.Dmelanogaster.UCSC.dm6.ensGene (v3.12.0) in combination with GenomicRanges (v1.42.0). Bam files were normalized using the bamCoverage function from deepTools with scale factors determined by the dba.normalize function from Diffbind. Consensus track files were generated between replicates using the mean function from the command line program wiggletools. Bandplot files for BED region subsets were generated using the computeMatrix and plotProfile functions from deepTools[90].

## ChIP-seq data analysis

To identify binding sites for CrebB, Hr38, and Sr, publicly available ChIP-seq data was obtained from the ENCODE project repository[61]. Specific files used for analysis were: CrebB (ENCFF090JJN, ENCFF655EMQ), Hr38 (ENCFF144OZH), Sr (ENCFF186BCY, ENCFF247KLE). ChIP-seq peaks for Hr38, Sr, and CrebB, were generated using optimal IDR thresholding by ENCODE[89], These peaks were annotated to the nearest gene using ChIPseeker (v1.26.2)[88]. Genome browser tracks were generated with pyGenomeTracks[91], using either control normalized or signal *p* value bigwig files generated by ENCODE.

## Cycloheximide feeding

To determine if translation is required during courtship memory formation, cycloheximide was fed to Nijmegen wild type male flies to block protein synthesis. Flies were fed media consisting of either 1% agarose, 5% sucrose alone (sucrose only) and with the addition of 35 mM cycloheximide (sucrose + CXM), as previously described[10]. Flies were raised on standard media and transferred to isolation chambers containing sucrose + CXM or sucrose-only 1 day prior to STM or LTM training. For STM, flies were fed sucrose + CXM during the 1-h rest period. For LTM, flies were fed either sucrose + CXM or sucrose-only media during the ~20-h rest period.

## Sample collection for single cell RNAseq

Virgin male flies carrying the CAMEL tool transgenes were collected every 4 h and allowed to mature individually in tubes for 3–4 days. *WT* and *Adcy1* mutant flies carrying the CAMEL tool were subjected to courtship LTM training and tested 20 h later, as described[40]. *WT* naïve flies carrying the CAMEL tool remained in the same tube without any female contact and were transferred to an empty test chamber. After testing, a capillary containing diluted yeast and sugar at the same concentration as standard food was placed in each test chamber and flies were left in an incubator for 2 h. After that, 40–50 flies were dissected for each condition and brain dissociation was performed as described[92]. In the case of *WTT* flies, we did an extra replicate.

## Fluorescence-activated cell sorting (FACS)

To FACS sort the CAMEL GFP positive cells from each condition we used an Influx cell sorter (Becton Dickinson) equipped with a 355 nm and 488 nm laser lines. We exclude aggregates using pulse processing and dead cells using DAPI as a viability dye. Cell were sorted directly into a 96-well plate with the lysis buffer to carry out single cell genomics.

## Single-cell cDNA and library preparation

cDNA was generated as described[93] with the following adjustments: preamplification of cDNA used 23 PCR cycles and was purified using Agencourt Ampure XP beads (Beckman Coulter) with a modified bead:DNA ratio of 0.8x. The quality of cDNA was checked using a NGS Fragment High Sensitivity Analysis Kit (Advanced Analytical) and a Fragment Analyzer (Advanced Analytical). The cDNA concentration was measured using a qubit high sensitivity dsDNA Kit. Libraries were prepared using a Nextera XT DNA Library Preparation Kit (Illumina), using a standard protocol but with all reaction volumes reduced by 1/10 to accommodate automation using the echo labcyte liquid handler (Beckman). Libraries were purified using Agencourt Ampure XP beads (Beckman Coulter). Size distribution of library pools was checked using a Fragment Analyzer and a NGS Fragment High Sensitivity Analysis Kit. Samples were pooled equimolar and the final pool quantified with the Kapa library quantification kit (Roche). The final pool was sequenced using the 150bp paired end kit on an Illumina NovaSeq 6000 with an average read depth of 10M reads per sample.

## Single Cell RNA seq data processing and analysis

FASTQ reads were quality checked using FastQC1 (v0.11.9) software and aligned against the *Drosophila melanogaster* reference genome release 6 (dm6) with STAR (v2.7.9) aligner. Htseq-count (v0.11.2) was then used to count the reads mapping each annotated feature. The obtained gene expression matrix was used as input to perform downstream analyses in Seurat v4.03. We first removed potentially lysed or bad quality cells by removing cells that showed high mitochondrial gene percentage (>5%) and genes detected in less than 3 cells, as well as cells with a different transcriptional profile from the most represented population. The remaining cells were then normalized, scaled and reduced using Principal Component Analysis (PCA). The data was then clustered via an optimization model of the K nearest neighbors algorithm (*k* = 15 or *K* = 25, depending on the analysis) using the *FindNeighbors* and *FindClusters* functions in Seurat. We performed

differential expression testing between clusters using the Wilcoxon rank sum statistical test. Differential marker gene lists were sorted by log2 Fold-Change, showing only those genes upregulated for each cluster and represented through heatmap plots using Complex-Heatmap R package. Removal of lobe-identity genes and the MB-enriched genes was performed directly on the Seurat object, with the same downstream processing as described above. DEGs from each single cell cluster were analyzed for GO and KEGG term enrichment using the R package ClusterProfiler, with FDR as the $p$ value correction method.

## Quantitative RT PCR

RNA was isolated from INTACT purified MB nuclei in triplicate using the PicoPure™ RNA Isolation Kit (Invitrogen: KIT0204) as per manufacturer's instructions. cDNA was synthesized from the isolated RNA using the SensiFAST™ cDNA Synthesis Kit (Meridian: BIO-65053) as per manufacturer's instructions. The cDNA was diluted 10-fold to be used for RT-qPCR using the SensiFAST™ SYBR No-ROX Kit (Meridian: BIO-98005) as per manufacturer's instructions. *Beta'COP* was used as the reference gene. The following primers were used:

*Beta'COP* (AACTACAACACCCTGGAGAAGG – ACATCTTCTCCC AATTCCAAAG)

*Hr38* (TGTCGCATCCCAACAGCAG - GAAGTGGCCGTGGTAGTT GTA)

*Sr* (AAGGGCTTGAAACCCTGGTG - CGAAGCTCAGCACATTGAA GTG)

## Staining and microscopy

To quantify CREB-positive MB cells, primary antibodies used were rabbit anti-GFP (1/200; Invitrogen ref. A11122) and mouse anti-Fasciclin II (1/50; DSHB AB_528235). To determine neurodevelopmental defects in the MB after knocking-down positive hits, we used anti-Fasciclin II (that identifies 'β' and γlobes) and mouse anti-Trio (1/10; DSHB AB_528494) (that marks α'β' and γ lobes). Images were taken using a Leica SP5 confocal microscopy, avoiding saturation with a 40X objective and with slices of 2.98 µm. Imaging of unc84::GFP was performed using a Zeiss LSM 900 using the primary antibodies; anti-GFP (1/300; Invitrogen G10362) and, as a marker for MB nuclei, anti-Dac (1/50; DSHB mAbdac1-1). Secondary antibodies used were Alexia 488 and 568 (1/500 or 1/300; Life Technologies).

CAMEL GFP positive neurons from 11-17 brains per condition were counted. Statistical significance was determined using an unpaired two tailed *t*-test ($p = 0.017$ for WT flies, $p = 0.75$ for *Adcy1* mutant flies). Between 9 and 19 brains per experimental condition were studied to detect MB defects in Fig. S9. Statistical significance calculated using the Fisher's exact test.

## Reporting summary

Further information on research design is available in the Nature Portfolio Reporting Summary linked to this article.

## Data availability

Raw and processed RNA-seq and ATAC-seq data generated in this study have been deposited in the GEO database under accession numbers GSE282414 and GSE274348. All raw single cell RNAseq data generated in this study have been deposited in the European Nucleotide Archive (ENA) under the Accession number PRJEB49180. The processed data are provided in Supplementary Data and Source Data files. Source data are provided with this paper.

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

## Acknowledgements

We thank the support of the Scientific Image and Microscopy, and the Omic Technologies and Bioinformatics facilities at Cajal Institute, the the Mount Saint Vincent University Bioimaging Facility, and Genome Quebec. We also appreciate flies and reagents from Dr. Jan Pielage, Dr. Gilbert Lee Henry, the Bloomington Drosophila Stock Center, the Vienna Drosophila Resource Center, the Developmental Studies Hybridoma Bank and the Transgenic RNAi Project at Harvard University. We thank the staff CNIO Flow Cytometry Core Unit team for their technical expertise optimizing and carrying out the flow experiments, in special Lola Martínez and Julia García-Lestón. This research was enabled in part by high performance computational infrastructure and training provided by the Atlantic Computational Excellence Network (https://www.ace-net.ca/) and the Digital Research Alliance of Canada (https://www.alliancecan.ca). We would like to acknowledge the invaluable work of our lab technicians Robert Reid-Taylor, Carmen Rodriguez-Navas and Esther Seco. Scripts used for processing courtship conditioning were adapted with the help of Nicholas Raun. Special thanks to our colleagues Prof Alberto Ferrús, Dr Sergio Casas-Tintó, Dr Pablo Méndez and Dr JL Trejo-Pérez for their helpful comments and suggestions on this manuscript. EASI-Genomics - This project has received funding from the European Union's Horizon 2020 research and innovation program under grant agreement No 824110. Part of the next-generation sequencing (NGS) data analysis was provided by the Genomics and NGS Core Facility at the Centro de Biología Molecular Severo Ochoa (CBMSO, CSIC-UAM) which is part of the CEI UAM + CSIC, Madrid, Spain-http://www.cbm.uam.es/genomica/. FAM was a recipient of a RyC-2014-14961 contract (2016-2022). Grant RyC-2014-14961 (FAM) funded by MICIU/AEI/10.13039/501100011033 and by ESF Investing in your future. Grant CNS2022-135223 (FAM) funded by MICIU/AEI/10.13039/501100011033 and by European Union NextGeneration EU/PRTR. Grant PID2022-142742NB-I00 (FAM) funded by MICIU/AEI/10.13039/501100011033 and by EDFR/EU. BG-M is a recipient of a FPI-UAM predoctoral fellowship, grant number SFPI/2020/00878. This project was also funded by a Project Grant (#363723) from the Canadian Institutes of Health Research to JMK and a Nova Scotia Graduate Scholarship to SGJ.

## Author contributions

F.A.M., J.M.K., and S.G.J. conceptualized and designed the project; J.M.K., F.A.M., and B.G.M. supervised the project; B.G.M. performed CAMEL scRNA-seq studies, S.G.J. performed INTACT RNA-seq and ATAC-seq. B.G.M. and J.M.K. performed confocal microscopy and B.G.M. analyzed the images. E.S.H. analyzed the scRNAseq data, S.G.J. analyzed INTACT RNA-seq and ATAC-seq data. B.G.M., S.G.J., E.B., N.M., A.C.E., T.B., and C.R. participated in the loss-of-function studies; S.G.J., B.G.M., E.T., T.B., J.M.K., and F.A.M. analyzed data; J.M.K. and F.A.M. wrote the original draft; S.G.J. and B.G.M. made the figures; E.F.B. performed qPCR. E.T. made extensive editing to the manuscript and revised statistics; all authors revised the manuscript; J.M.K. and F.A.M. were responsible for funding acquisition.

## Competing interests

The authors declare no competing interests.
