## [Transparent Peer Review file · Nature Communications]

A memory transcriptome time course reveals essential long-term memory transcription factors

Corresponding Author: Dr Francisco Martin

Version 0:

Reviewer comments:

Reviewer #1

(Remarks to the Author)

The study by Jones & Marti et al. offers a comprehensive investigation in *Drosophila* males aimed at identifying genes involved in long-term memory formation linked to sexual rejection. The authors conducted a comprehensive profiling of both whole heads (WH) and mushroom body (MB) nuclei following courtship rejection, followed by single-cell RNA sequencing (scRNA-seq) of CREB-activated MB neurons. By integrating a transcriptome time course, scRNA-seq, ATAC-seq, and a functional RNAi screen, they reveal a mushroom body specific transcriptional memory trace. They elucidate distinct phases of gene expression associated with learning and consolidation and identify novel candidate memory engram genes—most notably the immediate-early genes *Hr38* and *stripe*. The strengths of the work lie in its comprehensive approach, innovative use of temporal resolution, and the combination of functional and genomic analyses. However, weaknesses include potential limitations in the purity and validation of the isolated mushroom body nuclei, relatively low single-cell sampling that may underrepresent rare cell populations, reliance on embryonic ChIP-seq data to infer CREB binding in adult neurons, and concerns regarding RNAi specificity. I have included these issues, along with additional suggestions to strengthen the manuscript and widen its scope.

Major comments:

1. The study impressively combines a transcriptome time course, sc RNA-seq, and ATAC-seq to dissect the molecular underpinnings of LTM. However, the specificity and purity of the isolated MB nuclei purified by INTACT are not fully validated. Further independent validation (e.g., additional immunostaining or use of complementary cell-type markers) would strengthen the assertion that the observed transcriptional trace is truly MB-specific. In addition, how many copies of UAS does the *Unc84::2xGFP* carriers? Any ectopic expression from the *Unc84GFP* reporter should be quantified and mentioned.
2. Line 60-63 states “Several studies have performed transcriptome profiling of whole fly heads or MB neurons after memory training, which resulted in the identification of some genes that are induced by memory and required for memory formation^{25–32}. Despite this progress, there is little overlap observed between the different datasets.” While supplementary tables have some of this analysis, the main text doesn’t mention or describe this analysis properly. Also, several prior studies from this comparison are excluded e.g., dataset from Arbeitman lab and others. Any differences with pre-existing datasets and the potential reasons for variations should also be discussed.
3. The genes implicated in LTM are referred to as immediate early genes (IEGs), which is inaccurate, as they are also induced several hours post-training. A more appropriate term would be activity-regulated genes (ARGs), of which IEGs are only a subset. This distinction has been noted in previous literature, including by the Rosbash lab (Chen et al., *eLife*, 2016) and in mammalian studies from the Greenberg lab. Notably, *Hr38* and *Sr* have also been shown to respond to long-term social interactions between *Drosophila* males (Agrawal et al., *BMC Biol.*, 2019; Kent & Agrawal, *Mol. Neurobiol.*, 2020), but these findings were not referenced here. A comparative analysis using INTACT-purified dopaminergic neurons from this paradigm could provide additional insights into the generalizability of the findings to other paradigms.
4. The single-cell RNA-seq analysis is central to identifying putative memory engram clusters. Yet, the relatively low number of cells (48 naive, 37 *rut* mutants, 104 trained) raises concerns about statistical robustness and the potential for missing rare cell types. Increasing the number of profiled cells could have helped solidify the clustering and enhance reproducibility. However, in its absence, this limitation should at least be discussed.

5. The criteria for defining memory engram cells using the CAMEL reporter, while innovative, may need additional controls incorporating additional markers or dual-reporter strategies to rule out the possibility of contaminating non-engram neurons which would further refine the isolation of bona fide memory engram cells.

6. Line 761-764 states "To identify genes altered by memory training, differential expression analysis was performed between trained flies and time-of-day matched naive controls, using a cutoff of FDR < 0.1. To identify genes with enriched expression in the WH or MB, differential expression was performed between all MB and WH samples, using a cutoff of FDR <0.05 and log2 fold change > 0.5." What is the reason for these different thresholds? How the DEGs are affected of a Cutoff of FDR <0.05 is used throughout?

7. The criteria used for functional screening (lines 457–467) appear to be somewhat arbitrary, and a clearer justification for their selection is warranted.

8. The RNAi screen identifies several candidate genes affecting LTM which is important to establish causality. However, I am concerned about RNAi specificity and potential off-target effects, especially since UAS-dcr2 seem to have been used for these experiments. The description for the lof screen and GAL4 drivers, dcr2 etc. is buried in Methods and it should be mentioned clearly in main text and Figure 7. Was the Hr38 and Sr-RNAi Gal80ts experiments also included dcr2? If so, what happens without dcr2? Which of these RNAi lines are KK vs. TRiP vs. VDRC lines? How were the effect of white gene on courtship behavior mitigated given that KK lines have very high white expression? Incorporating rescue experiments for at least Hr-38 and Sr would help strengthen the findings.

9. A key mechanistic claim involves direct regulation by CREB, inferred from ChIP-seq data obtained from whole embryos. Given that CREB's binding landscape may differ substantially in adult MB neurons, performing ChIP-seq directly on MB nuclei from adult flies post-training would have provided more definitive evidence of direct regulation of immediate early genes (IEGs) such as Hr38 and stripe. In absence of this, the caveat should be appropriately discussed.

10. Most of the supplementary tables categorize the RNA-seq dataset into TIGs and TRGs, but the full dataset is not provided. Since these data represent a valuable resource for the field, the complete processed dataset (without FDR thresholding) for RNA-seq, ATAC-seq and scRNA-seq should be included as supplementary tables.

11. What is the reason for using a mixed genotype for parental line CS and Oregon-R? How does that affect baseline behaviors?

12. The study is focused solely on courtship conditioning as the paradigm for LTM formation. It remains unclear whether these transcriptional signatures and candidate regulators generalize to other memory paradigms (e.g., olfactory or aversive conditioning) or even other interaction paradigms (see my comment 3). Comparisons with existing studies across various paradigms would broaden the impact of the findings.

13. The manuscript is generally well written. However the discussion section would benefit from integrating a broader perspective by discussing how these findings relate to mammalian memory literature e.g., incorporating landmark studies on CREB and ARGs from Greenberg and Tonegawa labs and others might provide useful context and underscore the conservation of these mechanisms.

Minor comments:

1. The metabolic reprogramming observed in the MB—evidenced by sustained expression of energy metabolism genes—is intriguing. However, further discussion on how these changes mechanistically relate to synaptic remodeling and memory consolidation would be beneficial.

2. It would be useful to have 'read me' text in different supplementary tables describing various nonstandard abbreviations in headers (lfcSE, BgRatio etc.) and statistical analysis performed.

3. Genotype for parental line seem to have a typo 'Canton-R/Oregon-S'.

Reviewer #2

(Remarks to the Author)

The manuscript by Jones et al. investigates the transcriptomic dynamics associated with courtship long-term memory (LTM) formation and retrieval, focusing on CREB-activated neurons. The study provides valuable data that can advance our understanding of "transcriptional memory traces" and "transcriptional profiles for engram neurons," offering insights into the molecular underpinnings of LTM. However, the authors' claims require refinement or further experimental support. Memory traces likely extend beyond engram neurons, involving broader neural circuits where enduring synaptic and cellular plasticity underpin LTM storage, and neural output is essential for memory retrieval. To strengthen their conclusions, the authors should either provide direct evidence that the identified transcriptional changes are localized specifically to engram neurons or moderate their claims to reflect the possibility of broader circuit-level contributions.

Major Concerns:

1. R14H06-Gal4 Driver and MB Subtypes

The R14H06-Gal4 driver excludes α/β' Kenyon cells, omitting transcriptomic data from this critical MB subtype. While MB247-Gal4 is mentioned in the Methods, its application is unclear. Clarify whether MB247-Gal4 was used alongside anti-Fasciclin II and anti-Trio immunolabeling to differentiate MB subtypes, and discuss the implications of this limitation on the findings.

2. Differential Expression Analysis

RNA-seq was performed at eight time-points for trained flies but only five or six for naïve flies (inconsistent across text and Figure 1A). As gene expression also affected by circadian, it is essential to have both trained and naïve data from the same time point for the differential expression analysis.

3. RNA-seq Sampling and Memory Retrieval

RNA-seq analysis excludes the memory retrieval phase, although CAMEL labeling and RNAi experiments involve retrieval. This inconsistency is confusing. Including post-retrieval RNA-seq data would provide a more complete picture of transcriptional changes across memory phases.

4. Validation of CAMEL-Labeled Neurons

Confirm the functional relevance of CAMEL-labeled neurons by blocking their outputs (e.g., using Shibire or TNT) and testing their involvement in courtship LTM. This would validate their role as memory engram neurons.

5. Onset Sequence of CREB and Hr38 Transcription

Hr38 peaks within 60 minutes of activation, while CREB-driven transcription occurs later. To test if Hr38 upregulation results from retrieval-induced reactivation or sustained cycles, single-cell RNA-seq on CAMEL-GFP flies without retrieval is recommended.

6. RNAi Expression Timing for STM vs. LTM

Temporal GAL4 control via GAL80ts typically requires more than one day of high-temperature induction. In Figure 7, inconsistent RNAi induction times for STM and LTM assays are problematic. Uniform induction periods should be used to confirm RNAi effects.

7. Expression Timing of Hr38 and sr

Include STM training data to confirm that Hr38 and sr expression patterns differ under STM and LTM conditions, as Figure 8 lacks this comparison.

Minor Concerns:

1. Target Neurons and MB Involvement in Courtship Memory

While the mushroom body (MB) is well-established in olfactory memory, courtship memory integrates multisensory inputs and involves distinct unconditioned stimuli. To support the claim that the MB is the primary site of transcriptional changes in courtship memory, additional references and experimental evidence specific to courtship memory are required.

2. Hr38 and sr as Novel IEGs

The claim that Hr38 and sr are novel IEGs is overstated, as Hr38 is already recognized (Fujita et al., *Current Biology* 23.20, 2013). Revise this claim to reflect its established role while emphasizing its reactivation during retrieval.

3. Figure 1D Labeling

Label the black and yellow lines in Figure 1D to indicate the groups they represent.

4. Translation-Related Genes in MB Cells

In Figure 1E, translation-related genes are upregulated in whole-brain samples but not in the MB+MBWB groups. Clarify whether the upregulation of translation activity is specific to non-MB cells or if translation activity in MB cells remains consistently high, regardless of training.

5. Figure 3B and 3D Inconsistency

Clarify why Figure 3B shows no difference in CAMEL-positive cells between WT-trained and Rut-trained groups, while Figure 3D indicates Rut mutants fail to establish memory.

6. Clusters 1 & 2 in Figure 5B

The claim that clusters 1 and 2 represent early learning or consolidation is speculative. Provide evidence linking these clusters to memory processes, especially given their naïve and WTT-2 composition.

7. Training Induced Genes in Cluster A

Clarify the role of training induced genes in cluster A, as many cells in this cluster originate from the naïve group. Address potential circadian or environmental influences.

8. MB Ablation and Memory

Revise the statement to reflect the findings of de Belle and Heisenberg (*Science* 263.5147, 1994), which demonstrated that "MB ablation abolishes olfactory learning." The current phrasing, "MB ablation eliminates memory without drastically affecting learning or other behaviors," is inaccurate and should be corrected to align with the established literature.

9. Memory-Coded vs. Non-Memory-Coded KCs

Highlight transcriptional differences between memory-coded (WTT-1) and non-memory-coded (WTT-2) KCs. Emphasize metabolic regulation in WTT-2 and synaptic transmission upregulation in WTT-1 (clusters 4 and 5).

10. Reorganization of Supplemental Data

The supplemental data require reorganization to improve clarity and coherence. Additionally, numerous typos and instances of truncated text should be corrected to ensure the content is complete and accurate.

Reviewer #3

(Remarks to the Author)

This paper investigates the molecular and transcriptional mechanisms underlying memory formation in *Drosophila melanogaster*. By combining transcriptome time-course analysis, single-cell RNA sequencing and chromatin accessibility assays, specific changes to the mushroom body are analyzed. Two immediate early genes (Hr38 and stripe) are identified to play essential roles in LTM but not STM formation. Screening with RNAi confirms the role of these and other genes in

memory processes. The study further highlights chromatin remodeling associated with memory consolidation and in general provide a rich dataset which may facilitate the identification of memory engram cells.

The use of scRNA-seq, ATAC-seq, and RNAi screening provides a complete view of memory-related transcriptional processes and a functional validation of the role of the genes through behavior experiments. The study is strengthened by the use of innovative techniques such as INTACT for MB nuclei isolation and CAMEL for the identification of CREB-activated neurons.

The manuscript is interesting and particularly the study of *Drosophila* IEGs and their functional analysis of wide interest.

There are a few points that I feel are critical to be addressed:

The authors use an array of different techniques, most of which are converging on the transcriptional control of genes in learning conditions.

For the CAMEL sc-experiment however I see a couple caveats that should be addressed.

I) CAMEL is not specific for engram cells and there are typically many more cells that are labelled through the brain (in trained and untrained animals); thus it is not MB specific. The authors used whole-brains for isolation.

II) One would assume that the engram cells are CAMEL-positive after training, but “invisible” in non-trained animals or in animals lacking rutabaga. How can one compare CAMEL-positive cells with control conditions properly? Are there CAMEL-positive KCs in rut mutants? If so, it would argue for another pathway- if not, one is likely comparing CAMEL-positive KCs (in the experimental condition) with CAMEL-positive non- KCs (in the control condition), which would give molecular signature between the cell types.

- It would be interesting to know if, training the flies again, the transcripts that resulted to be upregulated in the first training are as well upregulated with the second, to better identify those genes that are surely required for memory encoding.

- Figure 1D,F,G: a legend on top of the figure would be helpful to rapidly understand the color coding of the lines, currently is not clear, especially for fig. 1D what does the red line represent? If it represent the naïve baseline, then the transcripts look downregulated rather than upregulated during memory formation (at least in the MB specific analysis and the MB and WH overlap)

- The transcriptomics experiments would offer more specific analysis. For instance, it would be important to know how known learning genes behave. This includes transcripts annotated for translation genes, G-protein and cAMP-related signaling pathways in trained flies which circadian rhythm has been disrupted.

Minor points:

Figure 2 is a bit dense, I might make sense to split into two figures

Line 2: Titles typically do not end with a period. Please modify

Line 32: the first mention of Hr38 should spell out the gene name as “Hormone receptor-like in 38”

Line 54 “lobesClick or tap here to enter text..”, please correct.

Version 1:

Reviewer comments:

Reviewer #1

(Remarks to the Author)

Thank you for addressing the comments and revising the manuscript. The revised manuscript is substantially improved but several critical issues remain that need to be addressed.

Major Comments

1. Thank you for clarifying the RNAi screen workflow and candidate selection; the added methodological details greatly improve transparency. However, I remain unconvinced by the argument against rescue experiments. In *Drosophila* it is routine to perform rescue experiments. Authors can use transgenes that are RNAi resistant by introducing silent mutations in the hairpin target site or by using a heterologous ortholog, so that the rescue transcript evades knockdown. Please include rescue of at least Hr38 or stripe with Gal80ts, to confirm that the observed LTM impairments result directly from loss of the target gene.

2. Thanks for performing additional comparisons and including the new data. However, Table S11 and Supplemental Figures S12-S13 are added only in the Discussion. Please also cite them in the Results section appropriately.

3. Line 305-314, The description of the 68-core TIGs across clusters 1-5 remains hard to follow and could mislead readers about which cell populations are involved. I recommend you rewrite this paragraph more schematically e.g., Cluster 3 shows low expression of the 68 TIGs, clusters 1-2 display strong induction consistent with early learning, and clusters 4-5 exhibit renewed expression indicative of recall activation.

4. The sentence “The CAMEL tool mainly expresses in $\alpha\beta$ and γ neurons, precisely the neuronal populations required for courtship LTM...” overstates exactness. Please tone down “precisely” to more measured term.

5. New confocal images support MB-nuclei purification (S1), but quantitative metrics are missing. Please mention that UAS construct carries 5X repeats in the main text.

6. The rationale for intersecting embryonic CREB ChIP with adult MB ATAC is reasonable, but a caveat is needed in Discussion acknowledging potential false negatives due to adult-specific binding sites not captured by embryo data. Also, the discussion section overstates these claims and implies definitive causality: "The binding of Hr38 and Sr to MB-specific TIGs and genes from the putative engram cells identified by scRNA-seq, suggests that they might contribute to the MB-specific transcriptional trace of courtship LTM." This needs to be toned down.
7. The discussion's single-sentence (line 567-569) acknowledgment of limited scRNA seq cell numbers is insufficient. Please expand this section by outlining how low sampling might miss rare engram subtypes.
8. Thanks for providing the full RNA seq and scRNA seq processed data. However, the full ATAC seq peak list is still absent.

Minor Comments

1. Line 352 "Bloomington Stock Drosophila Center" is misspelled.
2. Ensure that all axes, color keys, and cluster labels are consistent across main and supplemental figures.
3. The font sizes in several figure panels appear quite small and may not reproduce legibly in print. I recommend increasing axis labels and tick-mark labels and simplifying dense legends, so all text remains crisp and readable on paper.

Reviewer #2

(Remarks to the Author)

The author has substantially improved the manuscript by incorporating behavioral validation of CAMEL-labeled neurons in courtship long-term memory (LTM) formation and, more importantly, by reclassifying and reinterpreting the single-cell clustering data, which now offer a more coherent understanding. While not all concerns have been fully resolved, I believe the manuscript is ready for publication.

Reviewer #3

(Remarks to the Author)

The authors have done a terrific job with the revisions and addressed all points- I feel this is an excellent contribution to the field and I would like to congratulate them to their work!

Version 2:

Reviewer comments:

Reviewer #1

(Remarks to the Author)

The authors have done an excellent job addressing the concerns through thoughtful revisions and appropriate clarifications. While a rescue experiment would have strengthened the claims, their rationale for not including it is reasonable and the added clarification are satisfactory.

With the improvements made, I believe the manuscript is ready for publication and would like to congratulate the authors on a compelling and well-executed study.

Reviewer #1 (Remarks to the Author):

The study by Jones & Marti et al. offers a comprehensive investigation in *Drosophila* males aimed at identifying genes involved in long-term memory formation linked to sexual rejection. The authors conducted a comprehensive profiling of both whole heads (WH) and mushroom body (MB) nuclei following courtship rejection, followed by single-cell RNA sequencing (scRNA-seq) of CREB-activated MB neurons. By integrating a transcriptome time course, scRNA-seq, ATAC-seq, and a functional RNAi screen, they reveal a mushroom body specific transcriptional memory trace. They elucidate distinct phases of gene expression associated with learning and consolidation and identify novel candidate memory engram genes—most notably the immediate-early genes *Hr38* and *stripe*. The strengths of the work lie in its comprehensive approach, innovative use of temporal resolution, and the combination of functional and genomic analyses. However, weaknesses include potential limitations in the purity and validation of the isolated mushroom body nuclei, relatively low single-cell sampling that may underrepresent rare cell populations, reliance on embryonic ChIP-seq data to infer CREB binding in adult neurons, and concerns regarding RNAi specificity. I have included these issues, along with additional suggestions to strengthen the manuscript and widen its scope.

Major comments:

1. The study impressively combines a transcriptome time course, sc RNA-seq, and ATAC-seq to dissect the molecular underpinnings of LTM. However, the specificity and purity of the isolated MB nuclei purified by INTACT are not fully validated. Further independent validation (e.g., additional immunostaining or use of complementary cell-type markers) would strengthen the assertion that the observed transcriptional trace is truly MB-specific. In addition, how many copies of UAS does the *Unc84::2xGFP* carriers? Any ectopic expression from the *Unc84GFP* reporter should be quantified and mentioned.

In this study we performed INTACT using the 5XUAS-unc84-2XGFP driven by the R14H06-Gal4 driver line. To investigate the specificity of INTACT we have previously quantified the number of nuclei and percentage of GFP positive nuclei obtained using this system, which revealed at least a 90% specificity of GFP positive nuclei pulled down during INTACT (Jones et al 2018, G3). The R14H06-Gal4 driver is highly specific for MB gamma and alpha/beta neurons, with expression outside of the MB in only a few cells. This can be clearly observed at the fly light website <https://flweb.janelia.org/cgi-bin/flew.cgi> as well as in two previous publications where we published full confocal stacks of R14H06-Gal4 driving expression of UAS-mCD8-GFP (Chubak et al, 2019, DMM and Jones et al, 2018, G3). Here we provide additional images of *R14H06-Gal4* driving expression *5XUAS-unc84-2XGFP*, using *Dac* as a marker to label Kenyon cell nuclei (Fig S1). In these images, we see the high level of MB specificity, and also the few nuclei labeled outside of the MB. In addition, we tested for leaky expression in flies with 5xUAS-unc84-2XGFP and no Gal4 and observed no signal (Fig S1). Finally, our MB INTACT is validated by looking at the expression levels of key MB and non-MB genes in MB enriched INTACT samples compared to nuclei from whole head. For example, all widely used MB marker genes appeared significantly enriched, including *Adcy1 (rut)*, *Pde4 (dnc)*, *ey*, *prt*, *Dop1R1*, and

others. In addition, glia genes (e.g. *wrapper*), fat body genes (*Lsp2*), and eye genes (*ninaE/C/B/G*) are significantly depleted. The new images and a volcano plot highlighting key MB enriched and depleted genes are now included in the paper in a new supplemental Figure S1.

2. Line 60-63 states “Several studies have performed transcriptome profiling of whole fly heads or MB neurons after memory training, which resulted in the identification of some genes that are induced by memory and required for memory formation^{25–32}. Despite this progress, there is little overlap observed between the different datasets.” While supplementary tables have some of this analysis, the main text doesn’t mention or describe this analysis properly. Also, several prior studies from this comparison are excluded e.g., dataset from Arbeitman lab and others. Any differences with pre-existing datasets and the potential reasons for variations should also be discussed.

We agree with the reviewer. We re-wrote the whole discussion and included a comparison of our results with other published datasets, as well as discussing the potential reasons for variation among them. In the revised manuscript we now include a supplemental Table (Table S11) that contains the memory or social behaviour induced genes from 4 studies: 1) Agrawal et al 2019, BMC Biol., DAN neurons, $F_{\text{cross}} < 0.05$, $N=635$; 2) Winbush et al. 2012, G3, whole heads, $n=480$ induced transcripts; 3) Crocker et al. 2016, Cell reports, various neuron types, $p < 0.1$, $N=366$; 4) Widmer et al. 2018, elife, increased RNAPolIII occupancy in the MB by tissue specific DAM ID, $N=288$. We also added heatmaps of these genes showing their expression in our ABC single cell clusters and added these as new Supplemental Figures 12 and 13.

3. The genes implicated in LTM are referred to as immediate early genes (IEGs), which is inaccurate, as they are also induced several hours post-training. A more appropriate term would be activity-regulated genes (ARGs), of which IEGs are only a subset. This distinction has been noted in previous literature, including by the Rosbash lab (Chen et al., eLife, 2016) and in mammalian studies from the Greenberg lab. Notably, Hr38 and Sr have also been shown to respond to long-term social interactions between *Drosophila* males (Agrawal et al., BMC Biol., 2019; Kent & Agrawal, Mol. Neurobiol., 2020), but these findings were not referenced here. A comparative analysis using INTACT-purified dopaminergic neurons from this paradigm could provide additional insights into the generalizability of the findings to other paradigms.

We thank the reviewer for raising this point. We changed the IEG term to ARGs throughout the revised text. We also compared our results with published data from long-term socialization as well as with other studies regarding LTM (see above, point 2). We added this information in the introduction and in the revised discussion. We agree that the comparison to Agrawal et al is a valuable addition that we did not include in the original discussion.

4. The single-cell RNA-seq analysis is central to identifying putative memory engram clusters. Yet, the relatively low number of cells (48 naive, 37 rut mutants, 104 trained) raises concerns about statistical robustness and the potential for missing rare cell types. Increasing the number of

profiled cells could have helped solidify the clustering and enhance reproducibility. However, in its absence, this limitation should at least be discussed.

The reviewer is right; we also realized that the relatively low number of cells obtained in our study might cause us to miss underrepresented cell types or even clusters. We added a specific paragraph about this limitation in the discussion.

5. The criteria for defining memory engram cells using the CAMEL reporter, while innovative, may need additional controls incorporating additional markers or dual-reporter strategies to rule out the possibility of contaminating non-engram neurons which would further refine the isolation of bona fide memory engram cells.

We thank the reviewer for raising this important point. In response to this comment, as well as comments from the other reviewers, we have significantly altered the text and figures describing the single cell data. We agree that identification of specific bona fide engram cells is difficult, and potentially impossible. However, our data suggest that the comparison between the memory deficient *rut* mutant and the trained controls did reveal a transcriptional signature relevant to memory. For a detailed description of the changes and rationale, see the response to points I) and II) from Reviewer 3.

6. Line 761-764 states “To identify genes altered by memory training, differential expression analysis was performed between trained flies and time-of-day matched naive controls, using a cutoff of FDR < 0.1. To identify genes with enriched expression in the WH or MB, differential expression was performed between all MB and WH samples, using a cutoff of FDR < 0.05 and log₂ fold change > 0.5.” What is the reason for these different thresholds? How the DEGs are affected of a Cutoff of FDR < 0.05 is used throughout?

For DE analysis between Naive and Trained flies we collected n=3 replicates in most cases, and n=2 for some samples. Due to the low statistical power (n=3) and nature of the experiment (memory induced genes), we expect some changes to be subtle and on the threshold of statistical significance. Therefore, we used inclusive p-value cutoff of padj < 0.1. This cutoff is commonly used in the literature for this type of exploratory experiment. If we change the p-value cutoff here, we lose about 25% of DEGs, which is not desirable for our purposes. In contrast, when looking for MB enriched genes compared to WH we used all 38 samples for WH and MB, which gave us a lot of statistical power. The goal of this analysis was to identify genes that are highly enriched in the MB compared to WH. For this, we decided to use a restrictive cutoff of padj < 0.05 and an additional fold change cutoff log₂fc > +/-0.5. To explain our reasoning, we added the following text in the methods: “*To identify genes altered by memory training, differential expression analysis was performed between trained flies and time-of-day matched naive controls, using a cutoff of FDR < 0.1. To identify genes with enriched expression in the WH or MB, differential expression was performed between all MB (n=38) and WH (n=38) samples. With the large n values in this comparison, we used more stringent cutoffs (FDR < 0.05 and log₂ fold change > 0.5) to define MB enriched genes.*”

7. The criteria used for functional screening (lines 457–467) appear to be somewhat arbitrary, and a clearer justification for their selection is warranted.

We apologize for the lack of clarity. We used several defined criteria to select candidates for functional testing. We modified Figure S7 (previous Figure S5) where candidate genes are now highlighted and added a new Table S8 that summarizes the information used to select them. Additionally, we added several sentences in the main text of the Results section to clarify the criteria: "As a proof of causality, we compiled a list of candidate genes for functional testing. We first identified the 30 most differentially expressed genes among clusters A, B and C (Fig. S7A). Among these, we selected genes that i) showed high expression levels in most of the individual neurons within each cluster, ii) had a clear mammalian ortholog (at least a score of 4 via DIOPT v9.1 from Flybase), and iii) had a UAS-RNAi line available from the Bloomington Stock Drosophila Center (Fig S7A and Table S8). Genes with known roles in memory (such as *pkc53E* and *dnc*) or RNAi processing (such as *AGO1*) were discarded. We also filtered for additional Cluster C genes that showed high expression in at least 50% of cells, selecting additional genes from this group (Fig. S7B and Table S8). Since cells from all conditions (WTT, naive, and *Adcy1*) were present in cluster C, we also selected candidate genes that were highly expressed in more than 80% of WTT-1 neurons and in less than 20% of neurons from other conditions, favoring uncharacterized genes (annotated with CG numbers) (Fig. S7C and table S8)."

8. The RNAi screen identifies several candidate genes affecting LTM which is important to establish causality. However, I am concerned about RNAi specificity and potential off-target effects, especially since UAS-*dcr2* seem to have been used for these experiments. The description for the LOF screen and GAL4 drivers, *dcr2* etc. is buried in Methods and it should be mentioned clearly in main text and Figure 7. Was the Hr38 and Sr-RNAi Gal80ts experiments also included *dcr2*? If so, what happens without *dcr2*? Which of these RNAi lines are KK vs. TRiP vs. VDRC lines? How were the effect of white gene on courtship behavior mitigated given that KK lines have very high white expression? Incorporating rescue experiments for at least Hr-38 and Sr would help strengthen the findings.

We apologize that the description of these experiments is not clearer. Due to space restriction, we kept it minimal. All technical details of the crosses, including parental and test genotypes are available in Table S9. We also modified Figure S8 to contain detailed information on the genotypes of all crosses in the figure legend.

In summary, UAS-*dcr2* was used for some RNAi lines that contain long hairpin RNAs. These long hairpin RNAs are thought to be more efficient at knockdown with co-expression of *dcr2*. In total, 15 of 49 RNAi lines were tested using UAS-*dcr2*. Of these 8 demonstrated an LTM phenotype. Since ~50% of *dcr2* expressing lines did not demonstrate LTM defects and controls with *dcr2* expression showed normal memory, it appears that *dcr2* expression was not a major

cause of off-target memory defects. Notably, the RNAi constructs used contain no homology to any other gene, therefore minimizing the chances of off target knockdown. In several crosses (22 out of 49) we used *tubGal80(ts)* to avoid lethality and/or developmental defects. This implied that in these cases temperature was raised to 29°C. Importantly, we did not use *dcr2* for experiments in which this temperature shift was used. In previous work we had difficulties producing reliable memory responses in controls expressing *UAS-dcr2* in the MB at higher temperatures. Therefore, some long hairpin RNAs were used without *dcr2*. The raised temperature increases Gal4 induced gene expression, therefore potentially enhancing the knockdown potential of RNAi lines.

Experiments with *Sr* and *Hr38* used TRiP RNAi lines and did not use *UAS-dcr2*. For these genes we also tested a control with Gal4/*Gal80(ts)* and no RNAi, as well as a control containing the heterozygous RNAi transgene with no Gal4/*Gal80(ts)*. These all showed normal LTM. In addition, we validated the efficiency of the RNAi lines by crossing to *ActGal4* and observing lethal phenotypes. While a rescue experiment for *Hr38* and *Sr* would be interesting, we decided not to attempt this experiment. The main problem is that there is no suitable system to mimic the dynamic expression that we see in the normal scenario. *Hr38* and *Sr* are induced by memory training and subsequently gradually reduce their expression after several hours. Gal4 based rescue cannot achieve this, especially when Gal4 is being used to drive the RNAi. Even with an alternate expression system like LexA we do not know of a suitable tool to achieve the temporal resolution that we need. In addition, this experiment is complicated by the fact that the RNAi would knockdown the rescue expression construct as well as the endogenous gene. Despite this we believe that there is ample evidence suggesting no major off target effects for these lines. First, there is no homology between the RNAi construct and other genes. Second, results reported for *Sr* RNAi (BDSC 27701) and *Hr38* RNAi (BDSC 29376) on the TRiP website indicate no phenotype was observed with *dcr2/nanos-Gal4*, *eya composite-GAL4*, *pnr-Gal4*, *esg-Gal4*, *fru-Gal4*, *ato-Gal4*, *c587-gal4*, *Cg-Gal4*, *Pph13-GAL4*, and *Mef2-Gal4*. The lack of phenotypic effects with all these driver lines suggest that off target effect is not a major issue for these RNAi lines.

In this study, 6 out of 49 RNAi constructs were KK lines. None of these produced an LTM defect compared to the control. We did not control for white expression levels in these experiments; however, the controls and knockdown flies do have dark red eyes due to the presence of *Gal80(ts)* and *R14H06-Gal4* transgenes. The lack of any phenotype in these lines suggests that off-targets or non-specific effects due to white expression were not a confounding factor.

We feel that this information is now clear and accessible in the modified Table S9, and Figure S8. We also modified the text of the methods to make our strategies clearer. We did not add extensive explanation to the main text of the results because we are very near the word limit and because we feel it may add confusion to the reader. We would be happy to add more details in the main text if the reviewer feels this is essential.

9. A key mechanistic claim involves direct regulation by CREB, inferred from ChIP-seq data obtained from whole embryos. Given that CREB's binding landscape may differ substantially in

adult MB neurons, performing ChIP-seq directly on MB nuclei from adult flies post-training would have provided more definitive evidence of direct regulation of immediate early genes (IEGs) such as Hr38 and stripe. In absence of this, the caveat should be appropriately discussed.

Due to technical limitations, we are unable to do ChIP on transcription factors from INTACT isolated MB nuclei. We therefore overlapped published ChIP data for CREBB from embryos and overlapped those binding sites with MB-specific ATAC-seq peaks to identify binding sites that are in regions of MB open chromatin. Open chromatin, as measured by ATAC-seq, is one of the best predictors of tissue specific gene expression patterns (e.g. Natarajan et al., Genome Research, 2012). If embryonic CREB binding sites are present at regions of MB open chromatin, it is very likely that these represent MB binding sites too. It is possible that the MB has extra CREBB binding sites that are not detected in embryo, and these would be missed in our analysis. Therefore, our data may include false negatives, but our approach also has a strong advantage that it is not likely to produce false positives that might be generated if we did not exclude genes using MB-ATAC data.

10. Most of the supplementary tables categorize the RNA-seq dataset into TIGs and TRGs, but the full dataset is not provided. Since these data represent a valuable resource for the field, the complete processed dataset (without FDR thresholding) for RNA-seq, ATAC-seq and scRNA-seq should be included as supplementary tables.

For the RNAseq data, we modified Table S1 and S2 to include the complete differential expression analysis, instead of the filtered lists. Regarding scRNAseq, we added the normalized counts for each cell in the modified Table S5

11. What is the reason for using a mixed genotype for parental line CS and Oregon-R? How does that affect baseline behaviors?

This is a wild-type strain that was made over 15 years ago by JMK in Nijmegen, Netherlands. While attempting to do courtship conditioning experiments with wild type flies from Bloomington – Canton S and Oregon R - both of these strains showed very low courtship levels and were very problematic for large scale breeding. In general, the lines were unhealthy. JMK simply combined flies from both genotypes and allowed them to breed. The resulting wild type strain, which has been maintained in the lab ever since, has high courtship levels similar to outcrossed flies, and can be easily amplified to large numbers. The baseline behaviour of this strain is therefore more “normal” than that of the parent strains in Bloomington. We treat this line as our standard wild type. Based on the location where the flies were made, we now called this strain Nijmegen wild type.

12. The study is focused solely on courtship conditioning as the paradigm for LTM formation. It remains unclear whether these transcriptional signatures and candidate regulators generalize to other memory paradigms (e.g., olfactory or aversive conditioning) or even other interaction paradigms (see my comment 3). Comparisons with existing studies across various paradigms would broaden the impact of the findings.

We thank the reviewer for this suggestion. We compared our data with the lists of memory or socially induced genes from 4 studies (see above, point 2): We completely rewrote the discussion to highlight the overlapping genes between our study and others, as well as discussing possible reasons for the existing differences. We also added the data in Table S11 and Fig S12-13.

13. The manuscript is generally well written. However the discussion section would benefit from integrating a broader perspective by discussing how these findings relate to mammalian memory literature e.g., incorporating landmark studies on CREB and ARGs from Greenberg and Tonegawa labs and others might provide useful context and underscore the conservation of these mechanisms.

We added a paragraph in the new discussion in which we put our data in perspective together with the studies mentioned by the reviewer.

Minor comments:

1. The metabolic reprogramming observed in the MB—evidenced by sustained expression of energy metabolism genes—is intriguing. However, further discussion on how these changes mechanistically relate to synaptic remodeling and memory consolidation would be beneficial.

Several studies, in flies and mammals, have directly implicated energy metabolism in the process of LTM formation. In particular, we refer to a study from the group of Thomas Preat: “Placais et al, Upregulated energy metabolism in the Drosophila mushroom body is the trigger for long-term memory, 2017, Nat Comms”. In addition to this study, this group has published several follow up papers looking at the mechanisms of how glia contribute metabolites to the MB to help LTM formation. We also recently showed that energy metabolism is also essential for courtship LTM (Raun et al, PLoS Biology, 2025). We added a sentence to make the findings from Placais et al more clear: “*MB-specific energy metabolism via the TCA cycle is crucial for LTM, and not STM, in multiple Drosophila memory paradigms, including courtship LTM⁴¹ and aversive olfactory LTM⁴². After memory training the MB goes through a period of high pyruvate flux through the TCA. Increasing the ability of MB neurons to process pyruvate through the TCA allowed LTM consolidation to occur more easily, suggesting that ATP generation is a critical trigger for LTM formation⁴².*” While we would like to add a more detailed discussion of this point, we are very limited on space. If the reviewers feel that this is something we should expand on more, we would be happy to do so.

2. It would be useful to have ‘read me’ text in different supplementary tables describing various nonstandard abbreviations in headers (lfcSE, BgRatio etc.) and statistical analysis performed.

We apologize for this. We added a Column Descriptions tab within the supplementary tables S1, S2, S5 and S7.

3. Genotype for parental line seem to have a typo 'Canton-R/Oregon-S'.

We have replaced this genotype with "Nijmegen wild type".

Reviewer #2 (Remarks to the Author):

The manuscript by Jones et al. investigates the transcriptomic dynamics associated with courtship long-term memory (LTM) formation and retrieval, focusing on CREB-activated neurons. The study provides valuable data that can advance our understanding of "transcriptional memory traces" and "transcriptional profiles for engram neurons," offering insights into the molecular underpinnings of LTM. However, the authors' claims require refinement or further experimental support. Memory traces likely extend beyond engram neurons, involving broader neural circuits where enduring synaptic and cellular plasticity underpin LTM storage, and neural output is essential for memory retrieval. To strengthen their conclusions, the authors should either provide direct evidence that the identified transcriptional changes are localized specifically to engram neurons or moderate their claims to reflect the possibility of broader circuit-level contributions.

We agree with the reviewer's statement. We moderated our claims throughout the text concerning the engram cells. All three reviewers expressed similar concerns, which led us to reassess and reinterpret our data. We acknowledge that we may not be able to specifically identify courtship LTM engram cells with our approach. However, our data suggest that the comparison between the memory deficient *rut* mutant and the trained controls did reveal a transcriptional signature relevant to memory. For a detailed description of the changes and rationale, see the response to points I) and II) from Reviewer 3.

Major Concerns:

1. R14H06-Gal4 Driver and MB Subtypes. The R14H06-Gal4 driver excludes α'/β' Kenyon cells, omitting transcriptomic data from this critical MB subtype. While MB247-Gal4 is mentioned in the Methods, its application is unclear. Clarify whether MB247-Gal4 was used alongside anti-Fasciclin II and anti-Trio immunolabeling to differentiate MB subtypes, and discuss the implications of this limitation on the findings.

We apologize for the lack of information throughout the text. The complete genotypes and the crosses used are shown in Table S9, which we modified to improve clarity. All of the work here involved drivers that express in γ and α/β neurons, but not α'/β' . This includes MB247, R14H06-Gal4, and the CAMEL line (R21B06DBD). While a pan MB driver may have been interesting, most of them are expressed outside of the MB and in MB neuroblasts, for example the common MB driver OK107 (Aso et al, 2009 <https://doi.org/10.1080/01677060802471718>). Regarding MB247-Gal4, we only used this line in some of functional experiments shown in Fig

S8. In the original manuscript we had co-stained anti-TRIO and anti-FasII with the corresponding driver to determine whether there were major developmental defects in the MB. We added a co-staining of wild type MB247-Gal4, UAS-GFP with anti-TRIO and anti-FasII, showing that the driver expresses mainly in γ and α/β neurons. These images were added to Fig S9.

We chose to focus on these driver lines because of their high specificity for postmitotic MB neurons. In addition, our choice was guided by a series of studies from Krystyna Keleman's group that strongly implicate a recurrent circuit that involves MB gamma neurons in courtship memory. This recurrent circuit is activated during and after training and later strengthened during sleep to form long term memory courtship memory (Keleman et al 2012 doi: 10.1038/nature11345, Zhao et al, 2018 10.7554/eLife.31425, Kruttner et al 2015 10.1016/j.celrep.2015.05.037, Dag et al 2019 10.7554/eLife.42786). While α'/β' is implicated in courtship STM through silencing approaches such as Shi(ts) (Montague and Baker, 2016, 10.1371/journal.pone.0164516), their role in courtship LTM has not been explored, and they are not part of the known courtship LTM circuit involving the MB γ neurons.

2. Differential Expression Analysis

RNA-seq was performed at eight time-points for trained flies but only five or six for naive flies (inconsistent across text and Figure 1A). As gene expression also affected by circadian, it is essential to have both trained and naive data from the same time point for the differential expression analysis.

We performed RNA-seq at eight time points in trained flies and 5 for naive flies. As depicted in Figure 1A, we only performed differential expression analysis between naive and trained samples collected at the same time of day, with one exception. We used one naive sample, collected 8 hours after the start of training (8AST), to compare to the trained samples from both the 7AST and 8AST timepoints. We felt that samples collected within one hour, at midday, should be a suitable control for circadian effects. This strategy is clearly indicated in Figure 1A.

3. RNA-seq Sampling and Memory Retrieval

RNA-seq analysis excludes the memory retrieval phase, although CAMEL labeling and RNAi experiments involve retrieval. This inconsistency is confusing. Including post-retrieval RNA-seq data would provide a more complete picture of transcriptional changes across memory phases.

We agree with the reviewer that INTACT RNA-seq after recall could have been an interesting condition to collect. We reasoned that the long-lasting transcriptional changes involved in memory storage or recall should be more specific to engram cells. This is what we aimed to capture with CAMEL scRNAseq. While the broad transcriptional changes after memory recall may also be interesting, we felt that it was beyond the scope of this manuscript to address that experimentally.

4. Validation of CAMEL-Labeled Neurons

Confirm the functional relevance of CAMEL-labeled neurons by blocking their outputs (e.g., using Shibire or TNT) and testing their involvement in courtship LTM. This would validate their role as memory engram neurons.

We thank the reviewer for the suggestion. We performed the experiment by overexpressing TNT^G in CAMEL-labeled cells. Such neuronal inactivation blocked courtship LTM: flies did not significantly reduce their courtship behaviour 24h after training. This demonstrates that CAMEL positive cells are required for LTM and likely include some engram neurons. We added these results in Fig 4C, with associated modifications to the text.

5. Onset Sequence of CREB and Hr38 Transcription

Hr38 peaks within 60 minutes of activation, while CREB-driven transcription occurs later. To test if Hr38 upregulation results from retrieval-induced reactivation or sustained cycles, single-cell RNA-seq on CAMEL-GFP flies without retrieval is recommended.

The idea to perform scRNAseq on CAMEL cells with no recall is interesting. However, due to technical difficulties (batch effect) with combining scRNAseq experiments, time, and cost, we did not include this experiment. Because of this limitation in our experimental design, we cannot confidently distinguish re-activated cells from persistently activated engram cells. Therefore, we cannot know for sure if *Hr38* is induced in these cells persistently or as a response to retrieval. Based on similar comments from all reviewers we have extensively modified our interpretation and presentation of the scRNAseq data. Please see the response to Reviewer 3 point I) and II) for a detailed explanation.

6. RNAi Expression Timing for STM vs. LTM

Temporal GAL4 control via GAL80ts typically requires more than one day of high-temperature induction. In Figure 7, inconsistent RNAi induction times for STM and LTM assays are problematic. Uniform induction periods should be used to confirm RNAi effects.

***Hr38* and *sr* are rapidly induced by training. We hypothesized that this activated transcription is required for memory formation. Therefore, our goal with the RNAi knockdown was to limit the induction, not eliminate expression prior to induction. This is why we waited until one day before training to initiate the knockdown. However, to address the reviewers concern we now performed the temperature shift two days before STM training and found no defect in STM in *Sr* knockdown conditions, suggesting a specific role for *Sr* in LTM (Fig S8). In contrast, we observed not only STM defects for *Hr38*, but also reduced naive courtship, consistent with a broader role for *Hr38* in memory and social behavior (Fig. S8). This shows that *Hr38* has a role in the MB prior to memory training.**

7. Expression Timing of Hr38 and sr

Include STM training data to confirm that *Hr38* and *sr* expression patterns differ under STM and LTM conditions, as Figure 8 lacks this comparison.

To address this comment, we have performed INTACT followed by qPCR for *Hr38* and *sr* on LTM and STM trained flies at a time that corresponds to 1 hour after the end of LTM training. STM training induces no differences in *Hr38* and *Sr* expression at this timepoint, while clear induction in trained flies is observed after LTM training. This demonstrates differences in induction of these genes between STM and LTM conditions. This data was added to Figure 8.

Minor Concerns:

1. Target Neurons and MB Involvement in Courtship Memory

While the mushroom body (MB) is well-established in olfactory memory, courtship memory integrates multisensory inputs and involves distinct unconditioned stimuli. To support the claim that the MB is the primary site of transcriptional changes in courtship memory, additional references and experimental evidence specific to courtship memory are required.

We are not hoping to claim that the MB is the primary site of transcriptional changes for courtship memory. Likely other neurons also undergo important transcriptional changes. We chose the MB as a logical starting point based on an extensive body of literature that shows the importance of the MB in courtship STM and LTM. A series of studies from Krystyna Keleman's group strongly implicate a recurrent circuit that involves MB gamma neurons in courtship memory. This recurrent circuit is activated during and after training and later strengthened during sleep to form long term memory courtship memory (Keleman et al 2012 doi: 10.1038/nature11345, Zhao et al, 2018 10.7554/eLife.31425, Kruttner et al 2015 10.1016/j.celrep.2015.05.037, Dag et al 2019 10.7554/eLife.42786). In addition, McBride et al. demonstrated that MB ablation eliminates courtship STM and LTM, but does not affect courtship learning. We have added all these references to the introduction.

2. *Hr38* and *sr* as Novel IEGs

The claim that *Hr38* and *sr* are novel IEGs is overstated, as *Hr38* is already recognized (Fujita et al., Current Biology 23.20, 2013). Revise this claim to reflect its established role while emphasizing its reactivation during retrieval.

Indeed, *Hr38* and *sr* have both been identified as neuron activity responsive genes (ARGs). We did not mean to claim that this was novel, however, these genes have not previously been implicated as ARGs required during memory. This is a novel finding, however, we have moderated our claims and made sure to be clear that their role as general IEGs/ARGs was known.

3. Figure 1D Labeling

Label the black and yellow lines in Figure 1D to indicate the groups they represent.

We have improved our figure legend to make clear what the lines represent and have added a dashed line representing the naive baseline. We hope these changes improve the clarity.

4. Translation-Related Genes in MB Cells

In Figure 1E, translation-related genes are upregulated in whole-brain samples but not in the

MB+MBWB groups. Clarify whether the upregulation of translation activity is specific to non-MB cells or if translation activity in MB cells remains consistently high, regardless of training.

In Figure 1E, we see that whole head training induced genes are enriched for GO terms related to translation, while MB TIGs are not. This does not mean that the genes are not expressed in the MB; indeed, they are expressed at a moderate level, somewhat lower than in WH. The difference in GO enrichment reflects the fact that more translation genes are significantly induced in by training in the WH compared to the MB. As indicated in the text we found many translation genes both up and down regulated in WH and MB compared to naive flies. This prompted us to investigate the expression of translation genes as a whole; that data is presented in Figure 2A. This analysis led us to the conclusion that while translation genes are not largely induced in the MB, their expression dynamics is completely altered over time as a result of LTM training, and the expression peaks during consolidation. Clearly though, this is not specific to the MB, and it is logical that translation would be important also in the whole head during and after the social interactions experienced during training.

5. Figure 3B and 3D Inconsistency

Clarify why Figure 3B shows no difference in CAMEL-positive cells between WT-trained and Rut-trained groups, while Figure 3D indicates Rut mutants fail to establish memory.

Indeed, our data show that CAMEL positive cells are increased in rut-naive compared to WT-naive and there is no difference between rut-trained and WT-trained. Previously (Gil-Martí et al, *Scientific reports* 2024) we reported a similar trend in the number of CREB positive cells after social interaction in rut mutants, with more CAMEL positive cells in rut-naive than wt-naive flies. In controls, for both social and courtship conditioning, the output behavior in response to experience correlates with a difference in the number of CAMEL cells between the naive and trained conditions. This is something that did not happen in the case of *rut* mutant animals. The high variability seen in the case of rut mutant animals is probably because *rut(2080)* is a hypomorphic allele. Actually, some *rut(2080)* flies established memory (Fig 4E) and some *rut* mutant neurons clustered as engram cells, although less than from *wt trained* animals. Furthermore, many basal behaviors of *rut* mutant flies are different, such as aggression or sleep (Gil-Martí et al, *Scientific reports* 2024), so it is likely that the basal number of CAMEL-positive cells is increased, maybe because of a compensatory mechanism for CREB activation not involving *rut* as an adenylate cyclase.

6. Clusters 1 & 2 in Figure 5B

The claim that clusters 1 and 2 represent early learning or consolidation is speculative. Provide evidence linking these clusters to memory processes, especially given their naive and WTT-2 composition.

We thank the reviewer for raising this point. In fact, based on our new analysis we reclassified cluster 1 and 2 cells as non-MB neurons (see response to reviewer 3 point I and II). This claim was purely speculative, so we removed it.

7. Training Induced Genes in Cluster A

Clarify the role of training induced genes in cluster A, as many cells in this cluster originate from the naive group. Address potential circadian or environmental influences.

In the new analysis of our single cell data, we reclassified cluster A cells as non-MB neurons (see response to reviewer 3 point I and II). Therefore, we suggest that this transcriptional signature was related to experience-induced CREB-dependent activity in non-MB neurons. Despite not undergoing courtship LTM training, naive flies likely would undergo some experience dependent synaptic plasticity resulting in CREB activation through interactions with their environment. These non-MB CREB activated neurons likely reflect some of this incidental synaptic plasticity, and therefore it makes sense that they would have a similar transcriptional profile to CREB activated cells underlying courtship memory.

8. MB Ablation and Memory

Revise the statement to reflect the findings of de Belle and Heisenberg (Science 263.5147, 1994), which demonstrated that "MB ablation abolishes olfactory learning." The current phrasing, "MB ablation eliminates memory without drastically affecting learning or other behaviors," is inaccurate and should be corrected to align with the established literature.

Thank you for pointing out this mistake. We were referring to the results on courtship memory, where learning remains intact after MB ablation. We have corrected this statement.

9. Memory-Coded vs. Non-Memory-Coded KCs

Highlight transcriptional differences between memory-coded (WTT-1) and non-memory-coded (WTT-2) KCs. Emphasize metabolic regulation in WTT-2 and synaptic transmission upregulation in WTT-1 (clusters 4 and 5).

Given that WTT-2 were not KCs and of uncertain identity/function (see response to reviewer 3 point I and II), we removed most of these comparisons from the revised text.

10. Reorganization of Supplemental Data

The supplemental data require reorganization to improve clarity and coherence. Additionally, numerous typos and instances of truncated text should be corrected to ensure the content is complete and accurate.

We apologize for the lack of clarity, but this is due to the merged PDF of main text and tables, which changes the excel files. We have ensured that the revised supplemental information is complete and clear.

Reviewer #3 (Remarks to the Author):

This paper investigates the molecular and transcriptional mechanisms underlying memory formation in *Drosophila melanogaster*. By combining transcriptome time-course analysis, single-cell RNA sequencing and chromatin accessibility assays, specific changes to the mushroom body are analyzed. Two immediate early genes (*Hr38* and *stripe*) are identified to play essential roles in LTM but not STM formation. Screening with RNAi confirms the role of these and other genes in memory processes. The study further highlights chromatin remodeling associated with memory consolidation and in general provide a rich dataset which may facilitate the identification of memory engram cells.

The use of scRNA-seq, ATAC-seq, and RNAi screening provides a complete view of memory-related transcriptional processes and a functional validation of the role of the genes through behavior experiments. The study is strengthened by the use of innovative techniques such as INTACT for MB nuclei isolation and CAMEL for the identification of CREB-activated neurons.

The manuscript is interesting and particularly the study of *Drosophila* IEGs and their functional analysis of wide interest.

There are a few points that I feel are critical to be addressed:

The authors use an array of different techniques, most of which are converging on the transcriptional control of genes in learning conditions. For the CAMEL sc-experiment however I see a couple caveats that should be addressed.

1) CAMEL is not specific for engram cells and there are typically many more cells that are labelled through the brain (in trained and untrained animals); thus it is not MB specific. The authors used whole-brains for isolation.

We thank the reviewer for raising such a crucial point. Indeed, the reviewer was correct. The CAMEL tool is a split-Gal4 line with a 6x CREB binding sites (activation domain) plus the specific MB R21B06 driver (DNA-binding Domain), as described in Siegenthaler et al, 2019. We quantified the number of CAMEL positive neurons in the whole brain by confocal imaging (n=6) using anti-FasII as a specific $\alpha\beta$ and γ MB lobe marker in a naive flies. We observed that around 70% of neurons were located in the MB. We added two panels in Fig S3 (S3A, B) showing a representative brain with CAMEL cells and their quantification.

This new data forced us to reconsider our interpretation of single cell clusters. To more confidently identify KCs, we checked the expression of classical MB markers such as *eyeless*, *Dop1R2*, *oamb*, *mub*, *dac* and *pvt* in the WTT (reactivated memory) experimental condition, in which PCA clearly distinguished two populations (WTT-1 and WTT-2). These markers expressed in most WTT-1 but not in WTT-2 neurons, suggesting that only the former were proper MB cells. Previously we had classified the WTT-2 cluster (present in cluster 1,2, and A) as $\alpha'\beta'$ MB neurons due to the moderate presence of many general MB markers in many cells from all three experimental conditions (WTT, naïve, and *rut* mutant- see former Fig 3E), and because some of the most representative $\alpha'\beta'$ markers were also up-regulated in non-MB cluster 1-2 cells, such

as *trio* and *mamo* (see former fig 5C). However, our new interpretation is that cluster 1 and 2, and cluster A, represent non-MB neurons, while clusters 3-4 and 5 were MB $\alpha\beta$ and γ neurons, respectively. In the cell type independent cluster analysis, cluster B is enriched for the memory deficient *rut* mutant, while cluster C is enriched for WTT1, and likely represents a group of memory-relevant MB neurons.

II) One would assume that the engram cells are CAMEL-positive after training, but “invisible” in non-trained animals or in animals lacking rutabaga. How can one compare CAMEL-positive cells with control conditions properly? Are there CAMEL-positive KCs in *rut* mutants?

If so, it would argue for another pathway- if not, one is likely comparing CAMEL-positive KCs (in the experimental condition) with CAMEL-positive non- KCs (in the control condition), which would give molecular signature between the cell types.

These are all very pertinent questions and concerns, and similar comments were made by the other two reviewers. These comments helped us to realize that when we interpret our experiments, we need to consider that flies could be learning from their surroundings all the time and may also learn from the experiences that they have during the courtship conditioning protocol even when not paired with a mated female. Therefore, CAMEL positive neurons in the naive condition are to be expected, and all CAMEL positive neurons potentially represent some stage of memory formation, whether related to courtship LTM, the MB, or not. With that said we know the CAMEL cells are important for courtship LTM because we see them increase after training, and in a new experiment, we show that expression of TNT^G with the CAMEL tool eliminates courtship LTM (see response to reviewer 2, point 4).

Regarding the first question: "How can one compare CAMEL-positive cells with control conditions properly?". The reviewer makes an excellent point that it is difficult to find a suitable cell type to compare courtship LTM activated CAMEL cells. We cannot isolate the same cells from naive animals to legitimately compare the transcriptome before and after transition to an engram cell. Regarding the second question: "Are there CAMEL-positive KCs in *rut* mutants?". Yes, there absolutely are CAMEL positive KCs in the *rut* mutant condition and these cells might provide a suitable comparator. The *rut*²⁰⁸⁰ mutant allele we used is hypomorphic and we do see that some mutant flies established memory (Fig 4D). This probably correlates with the presence of *rut* mutant KCs in the memory-relevant Cluster C. Surprisingly, our quantitative data show that CAMEL positive cells are abnormally high in *rut* mutant naive animals. In controls, we observe an increase in CAMEL cells between naive and trained animals, but we do not see this in *rut* mutant animals. Rather, we see that CAMEL cells are high in *rut*-naive flies compared to WT-naive and there is no difference between *rut*-trained and WT-trained. Previously (Gil-Martí et al, *Scientific reports* 2024), we reported a similar trend in the number of CAMEL positive cells after social interaction in *rut* mutants, with more CAMEL positive cells in *rut*-naive than WT-naive flies. We do not understand why *rut* mutants show this unexpectedly high CAMEL signal in the naive state. It could be a compensatory mechanism through a *rut* independent pathway.

But importantly, we know that it is not involved in memory formation. Therefore, cluster B, containing mostly *rut* mutant cells, with also cells from naive and WTT1, likely represents CREB activated KCs that are not associated with memory. Therefore, Cluster B may represent a good control for non-memory CREB activated KCs. Since the ABC clustering is MB cell type independent, the different signatures are not likely to be due only to the signature expression between the cell types, but also likely reflect memory processes. This is supported by our functional screening data, where we identified a low percentage of genes from cluster B as positive hits, compared to Cluster C. Therefore, Cluster C, with the majority of cells from trained MB neurons, likely represents memory-relevant KCs, while Cluster B (mostly formed by *rut* mutant but also with presence of naive and WTT) might account for CREB-activated KCs not related to memory.

The new interpretation of our data prompted us to extensively modify the results and discussion and the accompanying new Figures 4, 5 and 6 (previous Figures 3, 4 and 5). We also modified the discussion accordingly.

- It would be interesting to know if, training the flies again, the transcripts that resulted to be upregulated in the first training are as well upregulated with the second, to better identify those genes that are surely required for memory encoding.

This is an interesting suggestion; however, this is not something that we can address directly. To our knowledge, a double training protocol has not previously been attempted with courtship memory. To address this comment, we would have to first establish a new paradigm, which is beyond the scope of this work.

- Figure 1D,F,G: a legend on top of the figure would be helpful to rapidly understand the color coding of the lines, currently is not clear, especially for fig. 1D what does the red line represent? If it represent the naive baseline, then the transcripts look downregulated rather than upregulated during memory formation (at least in the MB specific analysis and the MB and WH overlap)

We apologize for the lack of clarity with these plots. They show the relative induction in WH (black) and MB (orange) compared to the naive baseline (now added as a black dotted line). We now added better legends to these plots that clearly indicates this.

- The transcriptomics experiments would offer more specific analysis. For instance, it would be important to know how known learning genes behave. This includes transcripts annotated for translation genes, G-protein and cAMP-related signaling pathways in trained flies which circadian rhythm has been disrupted.

We have added a new supplementary Figure (Figure S2) with example plots of specific genes involved in the different pathways and processes that we discuss in the manuscript, including actin cytoskeleton, energy metabolism, translation and g-protein/cAMP signaling.

Minor points:

Figure 2 is a bit dense, I might make sense to split into two figures

Indeed, we split figure 2 into two figures.

Line 2: Titles typically do not end with a period. Please modify

Done

Line 32: the first mention of Hr38 should spell out the gene name as “Hormone receptor-like in 38”

Done

Line 54 “lobesClick or tap here to enter text..”,please correct.

Done

Reviewer #1 (Remarks to the Author):

Thank you for addressing the comments and revising the manuscript. The revised manuscript is substantially improved but several critical issues remain that need to be addressed.

We thank the reviewer for their thorough assessment of our work, and we agree that the changes we made resulted in an improved manuscript.

Major Comments

1. Thank you for clarifying the RNAi screen workflow and candidate selection; the added methodological details greatly improve transparency. However, I remain unconvinced by the argument against rescue experiments. In *Drosophila* it is routine to perform rescue experiments. Authors can use transgenes that are RNAi resistant by introducing silent mutations in the hairpin target site or by using a heterologous ortholog, so that the rescue transcript evades knockdown. Please include rescue of at least Hr38 or stripe with Gal80ts, to confirm that the observed LTM impairments result directly from loss of the target gene.

While it is routine to perform rescue experiments in *Drosophila* with null mutants, it is very uncommon when the gene depletion method is by RNAi. It is true that the Hassan group published a *Drosophila* RNAi escape strategy in 2009 (PMID: 19483100) and the Tomancak and Perrimon groups published papers about the use of cross species rescue in 2009 and 2010 (PMID: 19720858, PMID: 20126626). Despite these publications, we are not aware of any subsequent studies that use silent hairpin mutations or heterologous orthologs to rescue RNAi induced phenotypes. It is also standard practice to publish RNAi data without rescue (see example publications that also target Hr38 and/or Sr by RNAi - PMID: 38413599, PMID: 30967153). We also still believe that there is evidence that these RNAi lines have no off-target effects as stated in our original response, related to no homology of RNAi constructs and no obvious phenotype with several other Gal4 lines.

An additional point that was not addressed in our previous response is that the available UAS-Hr38 stock from FlyORF induces lethality when expressed with the wing driver MS1096 (PMID: 25568052). This adds an additional concern that correct expression levels are essential for a proper biological function, suggesting that the overexpression via Gal4 may have additional effects.

But from our point of view, the most important reason for not doing the rescue is that there is no suitable system to mimic the dynamic expression that we see in the normal scenario. We feel that this is an even more important problem than the issue of RNAi and cDNA co-expression. Indeed, lack of rescue might be inconclusive due to these technical caveats. In an effort to address the temporal function of Hr38 and sr, we are currently planning to develop new alleles with an inducible Kozak Gal4 insertion to mimic their expression patterns in an inducible null mutant. We are also examining regions of the promoters that might be used for new Gal4 lines. However, these tools require considerable time to create and validate, and we feel that this is beyond the scope of this manuscript.

Instead of performing additional experiments we have added cautionary statements to the revised manuscript and provide additional details about the RNAi lines including citing previous studies where they were expressed with no observed phenotype. For example:

Line 410 - “Our RNAi approach may include false negative results due to insufficient knockdown, or false positives due to off-target effects.”

Line 432 – “While off target effect are possible with RNAi, the lines used here have no homology to other genes and have shown no phenotype when expressed with several other tissue specific Gal4 drivers⁵⁴⁻⁶¹, indicating a low probability of off target effects.”

2.Thanks for performing additional comparisons and including the new data. However, Table S11 and Supplemental Figures S12-S13 are added only in the Discussion. Please also cite them in the Results section appropriately.

We have moved the comparison to the results section and modified the discussion accordingly to remove redundancy.

3. Line 305-314, The description of the 68-core TIGs across clusters 1-5 remains hard to follow and could mislead readers about which cell populations are involved. I recommend you rewrite this paragraph more schematically e.g., Cluster 3 shows low expression of the 68 TIGs, clusters 1-2 display strong induction consistent with early learning, and clusters 4-5 exhibit renewed expression indicative of recall activation.

Thanks for this suggestion, we have rewritten the section as described.

4. The sentence “The CAMEL tool mainly expresses in $\alpha\beta$ and γ neurons, precisely the neuronal populations required for courtship LTM...” overstates exactness. Please tone down “precisely” to more measured term.

Thanks for this suggestion, we have altered the sentence as follows:

“ The CAMEL tool mainly expresses in $\alpha\beta$ and γ neurons, which is similar to the expression domain of *R14H06-Gal4* and includes the neuronal populations required for courtship LTM^{36,37}.”

5. New confocal images support MB-nuclei purification (S1), but quantitative metrics are missing. Please mention that UAS construct carries 5X repeats in the main text.

Thank you for this comment, we have added the 5X repeats to the main text.

6. The rationale for intersecting embryonic CREB ChIP with adult MB ATAC is reasonable, but a caveat is needed in Discussion acknowledging potential false negatives due to adult-specific binding sites not captured by embryo data. Also, the discussion section overstates these claims

and implies definitive causality: “The binding of Hr38 and Sr to MB-specific TIGs and genes from the putative engram cells identified by scRNA-seq, suggests that they might contribute to the MB-specific transcriptional trace of courtship LTM.” This needs to be toned down.

Thank you for this comment. We have removed the indicated sentence from the discussion. We also added the following discussion of this caveat - Line 477: “If experimentally identified CHIP binding sites from embryo or pupae are present at regions of MB open chromatin, it is very likely that these represent MB binding sites too. It is possible that the MB has extra binding sites that are not detected in embryo, and these would be missed in our analysis. Therefore, our data may include false negatives, but our approach also has a strong advantage that it is not likely to produce false positives.”

7. The discussion’s single-sentence (line 567-569) acknowledgment of limited scRNA seq cell numbers is insufficient. Please expand this section by outlining how low sampling might miss rare engram subtypes.

Thanks for this comment, we have expanded this section as requested.

8. Thanks for providing the full RNA seq and scRNA seq processed data. However, the full ATAC seq peak list is still absent.

We have now added the ATAC-seq peak list as a new supplemental Table S10

Minor Comments

1. Line 352 “Bloomington Stock Drosophila Center” is misspelled.

Corrected

2. Ensure that all axes, color keys, and cluster labels are consistent across main and supplemental figures.

We have checked everything.

3. The font sizes in several figure panels appear quite small and may not reproduce legibly in print. I recommend increasing axis labels and tick-mark labels and simplifying dense legends, so all text remains crisp and readable on paper.

We have checked then figure text sizes and increased where appropriate.

Reviewer #2 (Remarks to the Author):

The author has substantially improved the manuscript by incorporating behavioral validation of CAMEL-labeled neurons in courtship long-term memory (LTM) formation and, more importantly, by reclassifying and reinterpreting the single-cell clustering data, which now offer a more coherent understanding. While not all concerns have been fully resolved, I believe the manuscript is ready for publication.

Reviewer #3 (Remarks to the Author):

The authors have done a terrific job with the revisions and addressed all points- I feel this is an excellent contribution to the field and I would like to congratulate them to their work!

We thank the reviewer 2 and 3 for their initial comments, and we agree that the new analysis and additional experiments we performed have substantially improved the paper.